# Robust Preference Optimization through Reward Model Distillation

**Adam Fisch**[*]                                                     *fisch@google.com*
*Google DeepMind*

**Jacob Eisenstein**[*]                                               *jeisenstein@google.com*
*Google DeepMind*

**Vicky Zayats**[*]                                                   *vzayats@google.com*
*Google DeepMind*

**Alekh Agarwal**                                                     *alekhagarwal@google.com*
*Google Research*

**Ahmad Beirami**                                                     *beirami@google.com*
*Google DeepMind*

**Chirag Nagpal**                                                     *chiragnagpal@google.com*
*Google Research*

**Peter Shaw**                                                        *petershaw@google.com*
*Google DeepMind*

**Jonathan Berant**[*]                                                *joberant@google.com*
*Google DeepMind*

**Reviewed on OpenReview:** *https://openreview.net/forum?id=E2zKNuwNDc*

## Abstract

Language model (LM) post-training (or alignment) involves maximizing a reward function that is derived from preference annotations. Direct Preference Optimization (DPO) is a popular offline alignment method that trains a policy directly on preference data without the need to train a reward model or apply reinforcement learning. However, the empirical evidence suggests that DPO typically assigns implicit rewards that overfit, and trend towards infinite magnitude. This frequently leads to degenerate policies, sometimes causing even the probabilities of the *preferred* generations to go to zero. In this work, we analyze this phenomenon and use *distillation* to get a better proxy for the true preference distribution over generation pairs: we train the LM such that its induced implicit reward, i.e., the scaled log-likelihood ratio of the model to the reference model, matches an explicit reward model trained on the preference data. Moreover, to account for uncertainty in the reward model we are distilling from, we optimize against a *family of reward models* that, as a whole, is likely to include at least one reasonable proxy for the preference distribution. Our results show that distilling from such a family of reward models leads to improved robustness to distribution shift in preference annotations, while preserving the simple supervised nature of DPO.

---

[*]Core contributor.

# 1 Introduction

Language model (LM) post-training (or alignment) aims to steer language model policies towards responses that agree with human preferences. Early state-of-the-art approaches have focused on reward learning from human feedback. In this paradigm, preference annotations are used to train reward models, which then guide the optimization of the language model policy through online reinforcement learning (an approach broadly referred to as RLHF). Recent research on offline "Direct Preference Optimization" (DPO; Rafailov et al., 2023) and extensions thereof (Azar et al., 2024; Tang et al., 2024b; Meng et al., 2024), however, has demonstrated that it is possible to directly optimize policies on the preference data, which (a) bypasses the need for a separate reward model, and (b) uses standard supervised techniques rather than online reinforcement learning, which can be more difficult to optimize. These advantages have led to the the adoption of offline alignment, and in particular offline DPO, as the post-training algorithm of choice in both smaller-scale academic settings as well as larger-scale projects such as Llama 3 (AI@Meta, 2024) and OLMo (Groeneveld et al., 2024).

While this direct approach to preference optimization is attractive in its simplicity and efficiency, it also raises questions about the effectiveness and robustness of the resulting policies—as well as the broader utility of an explicit reward model beyond online reinforcement learning. In this paper, we argue that explicit reward modeling can, in fact, offer substantial practical and theoretical benefits. In particular, we theoretically show that relying solely on the preference data can be a precarious strategy, with few natural brakes in place to prevent policies trained under the DPO objective from careening off towards degenerate policies when the preference data exhibits certain idiosyncratic properties. On the other hand, explicit reward models can easily be regularized and understood—regardless of whether they are Bradley-Terry models (Bradley and Terry, 1952), margin-based ranking models (Zhao et al., 2023), or any other function that correlates well with human preferences (Lee et al., 2023; Tang et al., 2024b; Swamy et al., 2024).

Taking a step back from pure direct preference optimization, we first explore a method that merges the best of both worlds for the offline setting: an efficient reward model distillation algorithm that (i) operates effectively in the offline setting, (ii) makes minimal assumptions about the true, optimal reward we aim to maximize, and (iii) demonstrates greater robustness to the specific distribution of prompt/response data used for policy alignment. Drawing inspiration from prior knowledge distillation techniques (Hinton et al., 2015; Romero et al., 2015; Yang et al., 2019; Furlanello et al., 2018), we use the same change of variables trick employed in DPO to express the language model policy in terms of its implicit reward model (Rafailov et al., 2023). We then train the policy's implicit reward model to match our desired, explicit reward via an $L_2$ loss that directly regresses the pairwise differences in target rewards for any two generation pairs $(x, y_1)$ and $(x, y_2)$.[1] Here we theoretically establish the equivalence between optimizing this simple distillation loss over a sufficiently diverse offline dataset of unlabeled examples, and optimizing the traditional online RLHF objective with reinforcement learning.

While this approach adds reward modeling and reward inference *back* into the pipeline, it still maintains much of the simplicity and efficiency of reward-model-free DPO. Specifically, in our setting, rewards for the training data can be computed offline, once, ahead of time. This computation is completely parallelizable, and reward inference is significantly faster than the autoregressive generation (also known as model rollout) that is done in online settings. Consequently, this allows the policy training framework to be nearly identical to standard DPO, modulo the structure of the data that is fed in. This is true regardless of the *size* of the policy that is trained (as the policy's implicit reward model does not need to be the same size as the explicit reward model) or its hyper-parameters (as the policy's hyper-parameters do not need to match those of the reward model).

Reward model distillation, however, is still subject to some of the same challenges facing DPO-style training. In particular, distillation requires having a reliable reward model—but having a reliable reward still requires having a reliable method for extracting a reward model from a potentially noisy preference dataset. To address the uncertainty surrounding what the "right" reward model to optimize against is, we also introduce a pessimistic extension to our approach. This extension aims to maximize the worst-case improvement of our model across a plausible family of reward models (e.g., those sufficiently consistent with annotated preference data). This strategy aligns with that of existing work in conservative offline reinforcement learning (Cheng

---

[1]We also note that a similar $L_2$ reward alignment loss was recently explored in independent work by Mao et al. (2024) and Gao et al. (2024b), giving further support for its effectiveness. See a discussion of related work in §2.

et al., 2022; Kumar et al., 2020). We show that this pessimistic objective can be equivalently expressed and optimized by adding a simple additional KL-divergence regularization to the original distillation objective.

Empirically, we find that reward model distillation, particularly pessimistic reward model distillation, leads to similar performance to prior direct preference optimization methods when the preference datasets used are unbiased. When the preference datasets are biased, however, it leads to significantly *better* performance when compared to DPO and the Identity Preference Optimization (IPO) framework of Azar et al. (2024), which was introduced as a more robust alternative to DPO. To further support these empirical observations, we provide an extensive theoretical analysis that both (i) sheds more light on the degenerative tendencies of DPO and issues inherent to its objective, and (ii) highlights relative advantages of our explicitly regularized approaches.

## 2 Related work

Recent work in offline alignment has focused on DPO (Rafailov et al., 2023) as a simpler alternative for aligning language models from preference data. Subsequent work, however, has identified issues with DPO, including weak regularization (Azar et al., 2024) and a tendency to decrease the probability of winning generations during training (Pal et al., 2024). Similar to some of the findings in this work, Rafailov et al. (2024a) theoretically showed the existence of minima to the empirical DPO loss that assign non-zero probability to outputs not present in the training data, which can lead to unexpected model behaviour. Here we further show that not only can such minima place non-zero probability on outputs not present in the training data, but they can also assign *near*-zero probability to *preferred* responses that do appear in the training data. A number of methods have since explored various avenues for combating these issues. These include analyzing the impact of noise on DPO alignment (Gao et al., 2024a), proposing to update the reference policy during training (Gorbatovski et al., 2024), and suggesting a variant of IPO with a per-context margin (Amini et al., 2024). Additional research has focused on token-level alignment methods (Zeng et al., 2024; Rafailov et al., 2024b; Mudgal et al., 2024; Chakraborty et al., 2024) and on developing a unified view of various offline alignment methods (Tang et al., 2024b). Similar to our contributions towards using pessimism with respect to the correct choice of reward model for robust reward model distillation, existing work on offline RLHF has also focused on encompassing various forms of conservative reward penalties (Zhu et al., 2023; Liu et al., 2024; Zhan et al., 2024). Most relevant to our work on a technical level, concurrent work by Mao et al. (2024) also uses an offline loss that leverages targets from a learned reward model for supervision (as opposed to binary preference labels) that has the same form of the simple "distillation loss" analyzed in §5.1 of this paper, but does not explore pessimism. Similarly, the REBEL algorithm of Gao et al. (2024b) also studies the same reward regression loss, but in the online setting. This work builds upon these findings, and provides further analysis, as well as a solution based on pessimism together with reward distillation.

As discussed in §1, offline settings are attractive since they allow for simple, efficient, and scalable training frameworks. At the same time, while offline alignment methods are popular, recent evidence suggests that online alignment methods such as RLHF (Christiano et al., 2017; Stiennon et al., 2020), can still lead to more favorable outcomes (Guo et al., 2024; Tajwar et al., 2024; Dong et al., 2024; Xu et al., 2024; Xiong et al., 2024; Calandriello et al., 2024), especially when high reward outputs have low probability under the base policy. One of the advantages of online settings over offline settings is that many strategies for mitigating over-optimization that are simple to apply in online settings, such as reward shaping, are not as straightforward to apply in offline settings. For example, reward ensembles have been widely investigated recently as a mechanism for tackling reward hacking in online RLHF (Eisenstein et al., 2023; Coste et al., 2023; Zhai et al., 2023; Ramé et al., 2024), and in the context of multi-objective optimization (Moskovitz et al., 2023; Rame et al., 2024). Other notable examples for combating reward over-fitting and optimization by training reward models with regularized training objectives include iterative data smoothing (Zhu et al., 2024), which uses a trained model to softly label data during RLHF, and reward calibration from demonstrations (Rita et al., 2024). This work addresses some of the methodological and experimental gap that exists between online and offline methods for RLHF, by allowing for explicitly designed, trained, and regularized reward models (or pessimistic reward model ensembles) to be added back into the *offline* alignment setting without losing the practical benefits of the offline setting. Also relevant to our work, Moskovitz et al. (2023) focus on "composite" rewards in the online setting, with the goal of achieving high task reward while ensuring that every individual

component is above some threshold—also by applying a Lagrangian relaxation. In this work, we also consider multiple reward models, but we only focus on cases where there is no known, obvious reward decomposition.

Finally, the question of using a small amount of offline data to learn high-quality policies, instead of online access to reward feedback, has also been widely studied in the offline reinforcement learning (RL) literature. The predominant approach here is to use pessimism, that is, to learn a policy with the highest reward under all plausible environment models consistent with the data, with an extensive theoretical (Liu et al., 2020; Zanette et al., 2021; Xie et al., 2021) and empirical (Kumar et al., 2020; Cheng et al., 2022; Yu et al., 2021) body of supporting work. The key insight in this literature is that without pessimism, the RL algorithm learns undesirable behaviors which are not explicitly ruled out in the training data, and pessimism provides a robust way of preventing such undesirable extrapolations, while still preserving generalization within the support of the data.

## 3 Preliminaries

We begin with a brief review of Direct Preference Optimization (DPO) (Rafailov et al., 2023).

### 3.1 The preference alignment problem

Let $x$ be an input prompt, $y \sim \pi_\theta(\cdot \mid x)$ be the language model policy $\pi_\theta$'s response to $x$, and $\pi_{\text{ref}}(y \mid x)$ a reference policy (such as a pretrained or finetuned language model that is high-performing, but not yet aligned, and often used as the starting point for optimization). Given some reward function $r^*(x, y)$, the goal of alignment is to solve for the "aligned" policy $\pi_{\theta^*}(y \mid x)$ that maximizes the following RLHF objective, i.e.,

$$\pi_{\theta^*}(y \mid x) = \arg\max_{\pi_\theta} \mathbb{E}_{\mu(x)} \left[ \mathbb{E}_{\pi_\theta(y|x)}[r^*(x, y)] - \beta \mathbb{D}_{\text{KL}}[\pi_\theta(\cdot \mid x) \| \pi_{\text{ref}}(\cdot \mid x)] \right], \tag{1}$$

where $\mu(x)$ is a fixed distribution over prompts, and the KL-divergence term keeps the aligned policy close to the anchoring reference policy, $\pi_{\text{ref}}(y \mid x)$. Here, the reward function $r^*$ is typically not known in advance, but rather inferred from collected human preference data in the form of $(x, y^w, y^\ell)$, where $x$ is the prompt, $y^w$ is the "winning", or preferred, response, and $y^\ell$ is the "losing", or dispreferred, response. A common approach is to assume that $(y_1, y_2)$ follow a Bradley-Terry model (Bradley and Terry, 1952), under which the probability that $y_1$ is preferred to $y_2$ given the reward function $r^*$ and prompt $x$ is $p^*(y_1 \succ y_2 \mid x) = \sigma(r^*(x, y_1) - r^*(x, y_2))$, where $\sigma(\cdot)$ is the sigmoid function and $\succ$ denotes preference. Under this model, we can use the preference data $(x, y^w, y^\ell) \sim \mathcal{D}_{\text{pref}}$ to estimate $r^*$ via maximum likelihood estimation, i.e.,

$$\hat{r} \in \arg\min_r \mathbb{E}_{(y^w, y^\ell, x) \sim \mathcal{D}_{\text{pref}}} \left[ -\log \sigma(r(x, y^w) - r(x, y^\ell)) \right]. \tag{2}$$

With $\hat{r}$ in hand, Eq. (1) can be optimized using standard reinforcement learning algorithms (Schulman et al., 2017; Stiennon et al., 2020; Christiano et al., 2017).

### 3.2 Direct preference optimization

DPO is a simple approach for offline policy optimization that uses preferences to directly align the language model policy, without training an intermediate reward model. Specifically, DPO leverages the fact that the optimal solution to the KL-constrained objective in (1) takes the form (Korbak et al., 2022)

$$\pi_{\theta^*}(y \mid x) = \frac{1}{Z(x)} \pi_{\text{ref}}(y \mid x) \exp\left(\frac{1}{\beta} r^*(x, y)\right), \tag{3}$$

where $Z(x) = \sum_y \pi_{\text{ref}}(y \mid x) \exp(\frac{1}{\beta} r^*(x, y))$ is the partition function. DPO reparameterizes the true reward function $r^*$ in terms of the optimal policy $\pi_{\theta^*}$ that it induces, i.e.,

$$r^*(x, y) = \beta \log \left( \frac{\pi_{\theta^*}(y \mid x)}{\pi_{\text{ref}}(y \mid x)} \right) + \beta \log Z(x). \tag{4}$$

Under the Bradley-Terry model, the likelihood that $y_1 \succ y_2$ can then be written as

$$p^*(y_1 \succ y_2 \mid x) = \sigma \left( \beta \log \frac{\pi_{\theta^*}(y_1 \mid x) \pi_{\text{ref}}(y_2 \mid x)}{\pi_{\theta^*}(y_2 \mid x) \pi_{\text{ref}}(y_1 \mid x)} \right), \tag{5}$$

where now $\pi_{\theta^*}$ can be directly estimated on $\mathcal{D}_{\text{pref}}$ following the objective in (2), in place of the intermediate reward model $\hat{r}$, i.e., $\pi_{\hat{\theta}}(y \mid x) \in \arg\min_{\pi_\theta} \mathcal{L}_{\text{dpo}}(\pi_\theta; \mathcal{D}_{\text{pref}})$ where

$$\mathcal{L}_{\text{dpo}}(\pi_\theta; \mathcal{D}_{\text{pref}}) = \mathbb{E}_{(y^w, y^\ell, x) \sim \mathcal{D}_{\text{pref}}} \left[ -\log \sigma \left( \beta \log \frac{\pi_\theta(y^w \mid x)\pi_{\text{ref}}(y^\ell \mid x)}{\pi_\theta(y^\ell \mid x)\pi_{\text{ref}}(y^w \mid x)} \right) \right]. \tag{6}$$

As described in §1, optimizing $\mathcal{L}_{\text{dpo}}$ offers two main advantages over using online RL for Eq. (1): (a) there is no need for a separate reward model, and (b) $\mathcal{L}_{\text{dpo}}$ is a supervised objective that can be trained offline, which allows for a simpler training setup than online learning. Still, $\mathcal{L}_{\text{dpo}}$ also has certain pitfalls, as we analyze next.

## 4 Pitfalls of direct preference optimization

As argued by Azar et al. (2024), DPO strongly relies on the Bradley-Terry assumption, which leads to surprising and undesirable consequences when trained on finite preference data. The root issue is that if we have any two responses $y_1$ and $y_2$ where $p^*(y_1 \succ y_2 \mid x) = 1$, then the Bradley-Terry model dictates that $r^*(y_1) - r^*(y_2) = +\infty$, and therefore $\pi_{\theta^*}(y_2 \mid x) = 0$ for *any* finite KL-regularization strength $\beta$.

We can illustrate this phenomenon on a broader level with the following example:

**Assumption 1.** *Suppose we are given a preference dataset of (context-free) pairs $\mathcal{D}_{\text{pref}} = \{(y_i^w, y_i^\ell)\}_{i=1}^n$, the pairs $(y_i^w, y_i^\ell)$ are mutually disjoint in both the elements. Further suppose that we optimize the DPO objective on $\mathcal{D}_{\text{pref}}$ with a single parameter $\theta_y$ for each $y$.*

**Proposition 1.** *Under Assumption 1, for any $(y, y')$ such that $y = y_i^w$ and $y' = y_i^\ell$ for some $i$, we have $\frac{\pi_{\theta^*}(y)\pi_{\text{ref}}(y')}{\pi_{\theta^*}(y')\pi_{\text{ref}}(y)} \to \infty$, for all global minimizers $\pi_{\theta^*}$ of the DPO objective in (6), for any $\beta > 0$.*

**Corollary 1.** *Under Assumption 1, further assume that $0 < \pi_{\text{ref}}(y) < 1$ for all $y$. Then $\pi_{\theta^*}$ is a global minimizer of the DPO objective in (6) iff $\pi_{\theta^*}(\mathcal{C}(y^\ell)^c) \to 1$ with $\pi_{\theta^*}(y_i^w) > 0 \ \forall i \in [n]$, where $\mathcal{C}(y^\ell)^c$ is the complement of the set of all responses $y$ that appear as a dispreferred $y_i^\ell$ for any $i \in [n]$.*

Additional analysis of the training dynamics of DPO is provided in §7. A significant implication of this result is that the set of global optima of the DPO loss includes policies that can shift nearly all probability mass to responses that never appear in the training set—and even assign near-zero probability to all of the training data responses that do in fact correspond to winning generations, $y^w$, a phenomenon that has been observed empirically and analyzed theoretically (Pal et al., 2024; Rafailov et al., 2024a;b; Tajwar et al., 2024).[2]

Stated differently, Corollary 1 implies that any $\theta^*$ merely satisfying $\pi_{\theta^*}(y_i^\ell) = 0$ with $\pi_{\theta^*}(y_i^w) > 0 \ \forall i \in [n]$ is a global minimizer of the DPO objective in this setting. Though simplistic, the scenario in Assumption 1 is closer to reality than might first be appreciated: in many practical situations we can expect the finite-sample preference data to contain one (or at most a few) preference annotations per example $(x, y_1, y_2)$, while the policies $\pi_\theta$ can have billions of parameters ($\gg n$). It is important to note that the supposed regularization term $\beta$ does not help: it can limit the speed at which the optimizer reaches the degenerate solution, but it cannot alter the final destination. This issue could be viewed as a classic instance of overfitting—but as opposed to *overpredicting* responses within the training set, we might overfit to *almost never* producing anything like the "good" responses that do appear within the training set. Furthermore, without additional regularization (beyond $\beta$), we can expect this degeneration to occur in typical preference datasets.

> **Takeaways: Pitfalls of direct preference optimization**
>
> The DPO objective only requires that the likelihood of the preferred response is *relatively* higher than that of the dispreferred response. A peculiarity of this objective is that when preference data is disjoint (i.e., preferred responses never appear as dispreferred responses, and vice versa), certain types of policies (e.g., over-parameterized) will learn to assign 0 probability to all dispreferred responses, with merely *non-zero* probability to all preferred responses. This includes policies that assign *near-zero* probability to preferred responses, and place all mass on (often degenerate) generations outside the training set.

---

[2]In fact, the same conclusions can also be inferred from Proposition 1 in the concurrent work of Rafailov et al. (2024a). While their assumptions are slightly different, similarly to us, they also target scenarios in which the training data exhibits certain (realistic) deficiencies.

### 4.1 Case study: degenerate DPO optima in a BoW model

For intuition on why the DPO global optima can include policies where $\pi(y^w)$ may be nearly 0 for all $y^w$ in the training set, consider the simplified case where the policy is a bag-of-words model, $\pi_\theta(y) \propto \exp(c(y) \cdot \theta)$ for $c(y)$ representing a vector of counts in $y$ and $\theta_i$ representing the unnormalized log-probability of token $i$. Then we can formally show that DPO optimization monotonically decreases an upper bound on the probability of the *preferred* completion, $\tilde{\pi}_{\theta^{(t-1)}}(y^w) \geq \tilde{\pi}_{\theta^{(t)}}(y^w) \geq \pi_{\theta^{(t)}}(y^w)$.

**Proposition 2.** *Let $y^w, y^\ell \in \mathcal{V}^n$ be preferred versus dispreferred outputs of length $n$, respectively, with $\pi_{\text{ref}}(y^w), \pi_{\text{ref}}(y^\ell) > 0$ and corresponding count vectors $c(y^w), c(y^\ell)$. Let $\log \pi_\theta(y) = c(y) \cdot \theta - nZ(\theta)$ for $Z(\theta) = \log \sum_i^\mathcal{V} e^{\theta_i}$, with upper bound $\log \tilde{\pi}_\theta(y) = c(y) \cdot \theta - n \max_j \theta_j$. Let $\theta^{(t)}$ represent the parameters of $\pi$ after $t$ steps of gradient descent on $\mathcal{L}_{\text{dpo}}(\{y^\ell, y^w, x\})$, with $\theta^{(0)} = 0$. Then, we have that $\pi_{\theta^{(t)}}(y^w) \leq \tilde{\pi}_{\theta^{(t)}}(y^w) \leq \tilde{\pi}_{\theta^{(t-1)}}(y^w)$ for all $t$, with strict inequality when $\|c(y^w) - c(y^\ell)\|_0 > 1$.*

If $\pi_{\theta^{(t)}}(y^w)$ decreases in $t$, what other strings become more probable? In the following proposition, we show that under the bag-of-words model, DPO optimization moves probability mass away from $y^w$ to sequences that contain only the tokens that maximize the difference between $y^w$ and $y^\ell$.

**Proposition 3.** *Let $y^w$ and $y^\ell$ be preferred versus dispreferred outputs of length $n$. Let $\Delta = c(y^w) - c(y^\ell)$ be the difference in unigram counts. Let $\hat{y} = [i, i, \ldots, i]$, for $i \in \arg\max \Delta$, with $\|c(\hat{y})\|_1 = n$. Then $\pi_{\theta^{(t)}}(y^w) - \pi_{\theta^{(t)}}(\hat{y}) = \tau(t)k$ for some $k \leq 0$ and some non-decreasing $\tau : \mathbb{Z}_+ \to \mathbb{R}_+$.*

We have $k = 0$ when $c(y^w) = c(\hat{y})$, and $k \ll 0$ when $\|c(y^w)\|_2 \ll \|c(\hat{y})\|_2 = n$ (when $c(y^w)$ is dense) and $\|\Delta\|_2 \approx \|\Delta\|_\infty$ (when $\Delta$ is sparse). This implies that when $y^w$ and $y^\ell$ are similar, $\pi_\theta(y^w)$ will degrade more rapidly. Early stopping will therefore have to trade off between reaching the degenerate solution on such cases, and underfitting other cases in which $y^w$ and $y^\ell$ are more distinct. Related findings are also reported in Razin et al. (2024).

> **Takeaways: BoW model case study**
>
> For a simple BoW model, we show a realized setting in which DPO causes the probabilities of preferred outputs to catastrophically plummet, and instead puts all probability mass on generations that simply maximize token-level differences between preferred and dispreferred examples.

## 5 Uncertainty-aware reward model distillation

As discussed in the previous section, a core issue in preference optimization is that the true preference distribution $p^*(y_1 \succ y_2 \mid x)$ is not known. Attempting to infer it from finite-sample preference data (which may further be biased or out-of-distribution with respect to the target domain) can then result in a failure to learn reasonable policies. In this section, we propose a regularized, pessimistic approach to direct preference optimization that brings explicit reward modeling back into the picture through a model distillation objective, while still maintaining the simplicity and efficiency of offline alignment methods.

### 5.1 Reward model distillation

Suppose for the moment that the reward function $r^*$ was in fact known, and did not have to be inferred from sampled preference data. Under this setting, we can then construct a straightforward and efficient offline optimization procedure that is similar in spirit to DPO, but no longer relies directly on a preference dataset. Concretely, given unlabeled samples $(x, y_1, y_2) \sim \rho$ (where the number of samples can be potentially unlimited), we can define a simple squared "distillation" loss that matches the pairwise differences of the explicit reward $r^*$ with those of the implicit policy reward defined by $\pi_\theta$, i.e.,

$$\mathcal{L}_{\text{distill}}(r^*, \pi_\theta; \rho) = \mathbb{E}_{\rho(x, y_1, y_2)} \left[ \left( r^*(x, y_1) - r^*(x, y_2) - \beta \log \frac{\pi_\theta(y_1 \mid x) \pi_{\text{ref}}(y_2 \mid x)}{\pi_\theta(y_2 \mid x) \pi_{\text{ref}}(y_1 \mid x)} \right)^2 \right]. \quad (7)$$

Due to symmetry, here it does not matter if $(y_1, y_2) = (y^w, y^\ell)$, i.e., preferred vs. dispreferred, or vice versa. Notably, a similar squared-loss objective has also been recently proposed and shown to be effective in

independent work by Mao et al. (2024) and Gao et al. (2024b). We now first provide further motivation for this approach, before extending it to pessimistic variants in the next section.

Intuitively, the distillation loss in (7) seeks to exactly match *differences* in reward model scores across all generation pairs $(x, y_1, y_2)$. It is easy to see that under the Bradley-Terry model, this is equivalent to matching the strength of the preference relationship, $y_1 \succ y_2$. Furthermore, by only matching differences, we can still conveniently ignore the log partition term, $\log Z(x)$, in the implicit reward formulation for $\pi_\theta$ as shown in (4), as it is constant across different $y$ for any given $x$. Finally, similar to the motivation in DPO, we can show that minimizing $\mathcal{L}_{\text{distill}}(r^*, \pi_\theta; \rho)$ indeed results in an optimally aligned policy $\pi_{\theta^*}$, as long as the data distribution $\rho$ has sufficient support over the space of prompts and responses.

**Theorem 1.** *Let $\mathcal{Y}$ denote the set of all possible responses for any model $\pi_\theta$. Assume that $\text{supp}(\pi_{\text{ref}}(y \mid x)) = \mathcal{Y}$, i.e., the reference policy may generate any outcome with non-zero probability. Further, let $\text{supp}(\rho(x, y_1, y_2)) = \text{supp}(\mu(x)) \times \mathcal{Y} \times \mathcal{Y}$. Let $\pi_{\theta^*}(y \mid x) \in \text{argmin}_{\pi_\theta} \mathcal{L}_{\text{distill}}(r^*, \pi_\theta; \rho)$ be a minimizer over all possible policies, of the implicit reward distillation loss in (7), for which $r^*(x, y)$ is assumed to be deterministic, and finite everywhere. Then for any $\beta > 0$, $\pi_{\theta^*}$ also maximizes the alignment objective in (1).*

The theorem holds for a broad class of data distributions $\rho(x, y_1, y_2)$, and makes no assumptions on $r^*$. For example, it is no longer necessary for it to be defined using a Bradley-Terry model. Notably, it applies for reward models that are much larger and potentially better than the policy model, as the reward model is not used at test time (in contrast to DPO, which ties the size of the policy model used for generation to the size of the implicit reward model that is trained on preferences). In fact, this result can also be seen as strict generalization of the IPO framework of Azar et al. (2024), which corresponds to the special case $r^*(x, y) \triangleq \mathbf{1}\{y = y_w\}$, if labeled pairs $(x, y_w, y_l)$ are provided instead of the unlabeled pairs $(x, y_1, y_2)$.

Of course, the true reward $r^*$ is usually not known in practice. Still, as in standard RLHF, we can construct good proxies by using the preference data to identify plausible target reward models $r_{\text{tgt}}$, further guided by any amount of regularization and inductive bias that we desire. Moreover, prior work has found that such explicitly trained reward models are more accurate and generalize better than the implicit reward model defined in (4) that is learned by DPO (Tang et al., 2024a; Lin et al., 2024). A natural choice is to thus first learn $r_{\text{tgt}}$ on the preference data $\mathcal{D}_{\text{pref}}$ using standard methods, and then reuse $\mathcal{D}_{\text{pref}}$ to distill $\pi_\theta$, which is similar to classical settings in teacher-based model distillation (Hinton et al., 2015; Romero et al., 2015). Furthermore, as $r_{\text{tgt}}$ is a real-valued model, at a bare minimum it is guaranteed to induce a regularized Bradley-Terry preference distribution $p_{\text{tgt}}(y_1 \succ y_2 \mid x) > 0$, $\forall x, y_1, y_2 \in \mathcal{X} \times \mathcal{Y} \times \mathcal{Y}$, and thereby avoid the degeneracies identified in §4 for the maximum likelihood estimate under DPO.

> **Takeaways: Reward model distillation**
>
> If the true reward $r^*$ is known, Eq. (7) provides an objective that distills $r^*$ directly into $\pi_{\theta^*}$, where $\pi_{\theta^*}$ maximizes $r^*$ as in Eq. (1). When the true reward $r^*$ is unknown, we can use *any form of approximate reward model* $r_{\text{tgt}}$. Furthermore, any real-valued $r_{\text{tgt}}$ naturally adds regularization, and avoids degenerate optima such as the 0-probability solutions identified in §4.

## 5.2 Pessimistic reward model distillation

Choosing a single reward model $r_{\text{tgt}}$ for anchoring the LM policy can naturally still lead to degenerate behavior if $r_{\text{tgt}}$ is a poor approximation of the true $r^*$ that accurately reflects human preferences. However, we can easily extend our framework to handle uncertainty in the right target reward function by defining a confidence *set* of $k \geq 1$ plausible target reward models, $\mathcal{S} = \{r_{\text{tgt}}^1, \ldots, r_{\text{tgt}}^k\}$, and training $\pi_{\theta^*}(y \mid x)$ to maximize the following "pessimistic" form of the objective in (1):

$$\max_{\pi_\theta} \min_{r_{\text{tgt}}^i \in \mathcal{S}} \mathbb{E}_{\mu(x)} \Big[ \underbrace{\mathbb{E}_{\pi_\theta(y|x)}[r_{\text{tgt}}^i(x, y)] - \mathbb{E}_{\pi_{\text{ref}}(y|x)}[r_{\text{tgt}}^i(x, y)]}_{\text{advantage over the baseline policy}} - \beta \mathbb{D}_{\text{KL}}(\pi_\theta(\cdot \mid x) \| \pi_{\text{ref}}(\cdot \mid x)) \Big]. \tag{8}$$

In this pessimistic objective we are no longer optimizing $\pi_\theta$ for a single reward, but optimizing $\pi_\theta$ to produce generations that are scored favorably on average, even by the worst-case reward model in the set $\mathcal{S}$, relative

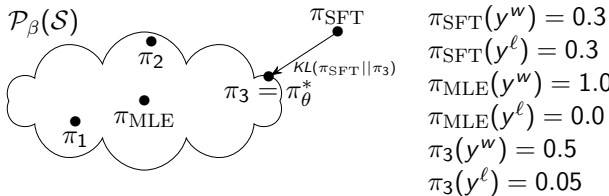

Figure 1: A toy illustration of Theorem 2, which states that the optimal $\pi_{\theta^*}$ for (8) is the policy in $\mathcal{P}_\beta(\mathcal{S})$ with the lowest forward-KL from $\pi_{\text{SFT}}$. The set $\mathcal{P}_\beta(\mathcal{S})$ contains a (potentially infinite) set of policies $\pi_1, \pi_2, \ldots$ corresponding to target reward models. Here, $\pi_{\text{SFT}}$ assigns equal mass to $y^w$ and $y^\ell$, $\pi_{\text{MLE}}$ is the MLE solution for the DPO objective, which puts all probability mass on $y^w$, and $\pi_3$ is the policy in $\mathcal{P}_\beta(\mathcal{S})$ with lowest forward-KL.

to the generations of the baseline policy $\pi_{\text{ref}}$.[3] When the set $\mathcal{S} = \{r^*\}$ consists of only the ground-truth reward, the objective (8) is equivalent to standard RLHF (1), up to a constant offset independent of $\theta$. More generally, whenever $\mathcal{S}$ includes a good proxy $\widetilde{r}$ for $r^*$, the pessimistic advantage evaluation ensures that the policy $\pi_\theta^*$ that maximizes eq. (8) still has a large advantage over $\pi_{\text{ref}}$ under all $r \in \mathcal{S}$, including $\widetilde{r}$. This use of pessimism to handle uncertainty in the knowledge of the true reward is related to similar techniques in the offline RL literature (Kumar et al., 2020; Cheng et al., 2022).

For the objective to be meaningful, however, the set $\mathcal{S}$ has to be chosen carefully. When $\mathcal{S}$ is small, it might not include any good proxy for $r^*$. Conversely, if $\mathcal{S}$ is too rich, it forces $\pi_{\theta^*}$ to be nearly identical to $\pi_{\text{ref}}$, since any deviations from $\pi_{\text{ref}}$ might be penalized by some reward model in $\mathcal{S}$. Consequently, we want to design $\mathcal{S}$ to be the smallest possible set which contains a reasonable approximation to $r^*$.

Finally, to solve (8), we can reformulate it as an equivalent constrained offline optimization problem, which conveniently admits a similar loss form as (7), as shown below:

**Theorem 2** (Pessimistic distillation). *Define the constrained minimizer*

$$\pi_{\theta^*}(y \mid x) \in \underset{\pi_\theta \in \mathcal{P}_\beta(\mathcal{S})}{\operatorname{argmin}} \ \beta \mathbb{E}_{\mu(x)} \mathbb{D}_{\text{KL}}(\pi_{\text{ref}}(\cdot \mid x) \| \pi_\theta(\cdot \mid x)), \tag{9}$$

*where $\mathcal{P}_\beta(\mathcal{S})$ is the set of all possible policies with implicit reward models that are consistent with any target reward model $r_{\text{tgt}}^i \in \mathcal{S}$, i.e., $\mathcal{P}_\beta(\mathcal{S}) \triangleq \{\pi_{\theta_i}\}_{i=1}^{|\mathcal{S}|}$ where $\pi_{\theta_i} \propto \pi_{\text{ref}}(y \mid x) \exp \frac{1}{\beta} r_{\text{tgt}}^i(x, y)$. Then for any $\beta > 0$, $\pi_{\theta^*}$ also maximizes the pessimistic alignment objective in (8).*

To unpack this result, Theorem 2 stipulates that the $\pi_\theta$ that maximizes the pessimistic objective in (8) is the policy in $\mathcal{P}_\beta(\mathcal{S})$ that is closest in *forward* KL-divergence to $\pi_{\text{ref}}$ (see Figure 1).[4] In addition, this policy also maximizes the expected reward of one of the $r_{\text{tgt}}^i \in \mathcal{S}$ (minus the additional weighted reverse KL-divergence penalty term). Intuitively, the forward KL-divergence term serves the role of biasing the model towards optimizing for reward models that are similar to the implicit reward that $\pi_{\text{ref}}$ already maximizes. Otherwise, there might exist a target reward model $r_{\text{tgt}}^i \in \mathcal{S}$ for which the advantage of $\pi_\theta$ relative to $\pi_{\text{ref}}$ will be low, or even negative (a solution that we would like to avoid).

### 5.2.1 Optimization

The constraint in (9) can be relaxed and approximately optimized by introducing an objective with a Lagrangian-style penalty with strength $\alpha > 0$ on a form of distillation loss as (7), i.e.,

$$\min_{\pi_\theta} \beta \mathbb{E}_{\mu(x)} \mathbb{D}_{\text{KL}}(\pi_{\text{ref}}(y \mid x) \| \pi_\theta(y \mid x)) + \alpha \min_{r_{\text{tgt}}^i \in \mathcal{S}} \mathcal{L}_{\text{distill}}(r_{\text{tgt}}^i, \pi_\theta; \rho), \tag{10}$$

---

[3]It is useful to optimize the *advantage* as it cancels the effects of constant differences between reward models in $\mathcal{S}$. We are also free to use any baseline policy $\pi_{\text{base}}$; we pick $\pi_{\text{ref}}$ for simplicity and ease of analysis in §7.

[4]Note that the objective in (9) minimizes the *forward* KL-divergence $\mathbb{D}_{\text{KL}}(\pi_{\text{ref}}(\cdot \mid x) \| \pi_\theta(\cdot \mid x))$ even though the pessimistic objective in (8) is regularized with *reverse* KL-divergence $\mathbb{D}_{\text{KL}}(\pi_\theta(\cdot \mid x) \| \pi_{\text{ref}}(\cdot \mid x))$.

where for convenience we divide by $\alpha$ and instead optimize[5]

$$\mathcal{L}_{\text{pdistill}}(\mathcal{S}, \pi_\theta; \rho) = \min_{r^i_{\text{tgt}} \in \mathcal{S}} \mathcal{L}_{\text{distill}}(r^i_{\text{tgt}}, \pi_\theta; \rho) + \gamma \mathbb{E}_{\mu(x)} \mathbb{D}_{\text{KL}}(\pi_{\text{ref}}(\cdot \mid x) \| \pi_\theta(\cdot \mid x)), \tag{11}$$

where $\gamma = \beta/\alpha$. In reality, minimizing (11) for $\gamma > 0$ is equivalent to solving the constrained optimization problem in (9) with an implicitly larger set of possible reward models $\mathcal{S}_\gamma \supseteq \mathcal{S}$ indexed by $\gamma$. More specifically, $\mathcal{S}_\gamma$ also contains all reward models $\tilde{r}$ that are approximately consistent with the anchoring reward models $r^i_{\text{tgt}}$ contained in $\mathcal{S}$, as the following result states.

**Proposition 4** (Soft pessimistic distillation)**.** *Assume the same conditions as Theorem 1. Then for any $0 < \gamma < \infty$, there exists a $\lambda \geq 0$ such that $\pi_{\theta^*}(y \mid x) \in \arg\min_{\pi_\theta} \mathcal{L}_{\text{pdistill}}(\mathcal{S}, \pi_\theta; \rho)$, where $\pi_{\theta^*}$ is a minimizer over all possible policies of the objective (9), for the effective reward model set*

$$\mathcal{S}_\gamma = \bigcup_{r^i_{\text{tgt}} \in \mathcal{S}} \left\{ \tilde{r} \colon \mathbb{E}_{\rho(x, y_1, y_2)} \left[ (r^i_{\text{tgt}}(x, y_1) - r^i_{\text{tgt}}(x, y_2) - \tilde{r}(x, y_1) + \tilde{r}(x, y_2))^2 \right] \leq \lambda \right\}. \tag{12}$$

As a result, optimizing (11) even when using the singleton $\mathcal{S} = \{r_{\text{tgt}}\}$ yields an implicitly pessimistic objective, in which the pessimism is over all reward models $\tilde{r}$ that are consistent up to $\lambda$ with $r_{\text{tgt}}$.

> **Takeaways: Pessimistic reward model distillation**
>
> It can often be unclear what reward model should be used as a target for policy distillation. A natural optimization objective in the face of uncertainty is to optimize for the *worst-case* reward, out of a set of plausible reward candidates. We show that this can be formulated in a very similar style to the original distillation loss in Eq. (7) by simply adding an additional forward KL penalty term.

## 5.3 Pessimistic DPO

Proposition 4 can also be leveraged to obtain an alternative, implicitly pessimistic, objective that uses DPO directly instead of distillation. Consider the following regularized DPO loss:

$$\mathcal{L}_{\text{pdpo}}(\pi_\theta; \mathcal{D}_{\text{pref}}) = \mathcal{L}_{\text{dpo}}(\pi_\theta; \mathcal{D}_{\text{pref}}) + \gamma \mathbb{E}_{\mu(x)} \mathbb{D}_{\text{KL}}(\pi_{\text{ref}}(y \mid x) \| \pi_\theta(y \mid x)). \tag{13}$$

Following a similar analysis as in Proposition 4, we can derive that this implicitly corresponds to maximizing the pessimistic objective in (8) for the reward model set

$$\mathcal{S}_\gamma = \left\{ r_{\pi_\theta} \colon \mathcal{L}_{\text{dpo}}(\pi_\theta; \mathcal{D}_{\text{pref}}) \leq \min_{\pi'_\theta} \mathcal{L}_{\text{dpo}}(\pi'_\theta; \mathcal{D}_{\text{pref}}) + \lambda \right\}, \tag{14}$$

where $r_{\pi_\theta}(x, y) \triangleq \beta \log \pi_\theta(y \mid x)/\pi_{\text{ref}}(y \mid x) + \beta \log Z(x)$ is the implicit reward model defined by $\pi_\theta$. $\mathcal{S}_\gamma$ then corresponds to the set of reward models $r_{\pi_\theta}$ that are all approximate minimizers of the DPO loss. This includes not only the MLE, but also all other estimators that obtain nearly the same loss. In principle, this can be expected to help ameliorate some of the issues of §4: since driving the reward to $\pm\infty$ only marginally decreases the $\mathcal{L}_{\text{dpo}}$ loss past a certain point, the set $\mathcal{S}$ will also include finite reward functions $|r_{\pi_\theta}(x, y)| < \infty$ for any $\gamma > 0$. These rewards would then be preferred if they induce a policy with a smaller (forward) KL-divergence to $\pi_{\text{ref}}$ than the degenerate, infinite rewards.

## 6 Experimental results

The main motivation for reward distillation and pessimism is to increase alignment robustness in challenging settings where it is difficult to learn good policies directly from the preference data. To demonstrate the effectiveness of our approach, we run experiments on the popular TL;DR summarization task (Stiennon et al., 2020; Völske et al., 2017), in which we simulate a scenario where the preference data has a spurious correlation between the *length* of a summary and whether or not it is preferred.[6] Additionally, we show results for an *unbiased* setting on TL;DR, as well for an unbiased setting on Anthropic Helpfulness (Bai et al., 2022).

---

[5]In practice, we also compute and optimize the min over reward models per each mini-batch of examples.

[6]Length has been repeatedly shown in the past to correlate with reward (Singhal et al., 2023; Park et al., 2024).

### 6.1 Experimental setup

We first train an "oracle" reward model on the TL;DR preference data training set (Stiennon et al., 2020) and relabel all preference pairs with this oracle. This enables us to use the oracle reward model for evaluation, without worrying about the gap to true human preferences. After relabeling, longer responses (where longer is defined as $y_1$ having at least 10% more tokens than $y_2$) are preferred in 61% of the examples.

To test the effect of a spurious correlation on preference-based policy optimization, we select a training set of 30K examples from the relabeled data such that the longer output is preferred in $\rho$ fraction of examples, with $\rho \in \{0.2, 0.3, 0.4, 0.5, 0.6, 0.7, 0.8\}$. Each such training set is denoted $\mathcal{D}_\rho$. At each $\mathcal{D}_\rho$, we compare our approach to DPO (Rafailov et al., 2023) and IPO (Azar et al., 2024), which are currently the most commonly used offline alignment methods. We test the following variants of distillation and pessimism:

- **Distilled DPO** (d-DPO): Trains a reward model $r_\rho$ on $\mathcal{D}_\rho$, and then optimizes $\mathcal{L}_{\text{distill}}(r_\rho, \pi_\theta; \rho)$.

- **Pessimistic DPO** (p-DPO): A pessimistic version of DPO as described in §5.3, trained on $\mathcal{D}_\rho$.

- **Pessimistic Distilled DPO** (dp-DPO): Combines the above two by training a reward model $r_\rho$ on $\mathcal{D}_\rho$ and optimizing the pessimistic distillation objective (Eq. (11)) with confidence set $\mathcal{S} = \{r_{\text{tgt}}\}$.

- **Pessimistic Ensemble DPO** (e-DPO): To create ensembles of reward models, we subsample from each $\mathcal{D}_\rho$ five preference datasets, $\mathcal{D}_{\rho,b}$, at $b \in \mathcal{B} = \{0.2, 0.4, 0.5, 0.6, 0.8\}$, such that the fraction of pairs where the longer response is preferred is $b$, and train reward models $r_{\rho,b}$ on those subsets. Consequently, sensitivity to length should vary across ensemble members. We then apply the same procedure as dp-DPO above, with a confidence set $\mathcal{S}_\rho = \{r_{\rho,b}\}_{b=1}^{\mathcal{B}}$.

All reward models and policies are initialized from Palm-2-XS (Anil et al., 2023). Policies also go through a supervised finetuning step on human-written summaries from the original TL;DR training set (Völske et al., 2017) prior to alignment, and we term this policy $\pi_{\text{SFT}}$. We evaluate performance by sampling summaries for test set prompts, evaluating the average reward according to the oracle reward model, and computing the advantage in average reward compared to $\pi_{\text{SFT}}$ (before alignment). We train policies for $10{,}000$ steps with batch size 16 and learning rate $10^{-6}$, and reward models for $3k$ steps with batch size 64 and learning rate $4 \times 10^{-6}$. We use the validation set for model selection during policy training and to choose the following hyperparameters. For all DPO variants, we sweep over $\beta \in \{.01, .1, 1, 3, 10, 30, 100\}$. For IPO, we sweep over $\tau \in \{0.01, 0.1, 1, 3, 5, 10, 25\}$. For all pessimistic methods we anneal $\gamma = \alpha/\beta$ from $10^{-4}$ to $10^{-2}$ linearly during the $10k$ training steps (however, in later experiments performed with e-DPO, we found annealing does not affect performance and a constant $\gamma$ also leads to similar performance, see Figure B.5).

### 6.2 Results

We present the results of our experiment in Figure 2. As can be seen in the plot, the more challenging setting is when $\rho < 0.5$, which corresponds to a sample of preference annotations in which shorter outputs are generally preferred. This distribution shift is more difficult because as mentioned the oracle reward model (trained on human annotations) has a bias in favor of longer outputs (Singhal et al., 2023). Nevertheless we get sizable improvements compared to the reference policy $\pi_{\text{SFT}}$ for all length bias values.

All approaches that invoke distillation (d-DPO, e-DPO, dp-DPO) outperform IPO and DPO ($p < .01$ by a Wald test) for $\rho \leq 0.5$, where shorter responses are preferred. Pessimistic ensemble DPO (e-DPO) performs particularly well in these settings, generally outperforming all methods that use a single reward model. When longer responses are preferred ($\rho > 0.6$), single reward distillation (d-DPO) leads to the highest performance, significantly outperforming both DPO and IPO ($p < .01$ by a Wald test). Interestingly, p-DPO does not provide empirical benefits relative to the distillation based methods, indicating that the distillation loss itself is quite important. For the effect of hyper-parameter selection, see Figure B.4. In DPO-based methods, the optimal value of $\beta$ is inversely correlated with the bias; in IPO the same holds for the $\tau$ hyperparameter.

To better understand the utility of reward ensembles in e-DPO, in particular when $\rho < 0.5$, we examine the role of each reward model in the ensemble across different biases. Specifically, for e-DPO, we identify for each example, throughout training, the reward model $r_{\rho,b}$ that best matches the implicit reward of the current

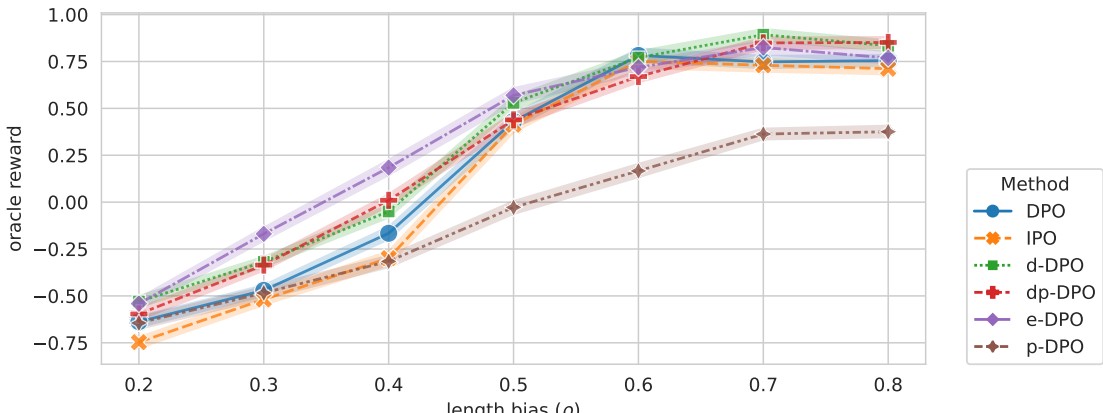

Figure 2: **Main results**, showing the oracle reward compared to the initial finetuned policy (the oracle reward of the initial finetuned policy is $\approx -1$). Error bars correspond to bootstrap 95% confidence intervals for finite sample variance. Ensemble DPO (e-DPO) is significantly better than DPO and IPO in the challenging setup where shorter responses are preferred ($\rho \leq 0.5$), and is generally the best-performing method overall in this regime. Distilled DPO (d-DPO) performs best when longer responses are preferred ($\rho > 0.6$).

policy, i.e., for which reward model is $\mathcal{L}_{\text{distill}}$ minimized on that example (see Eq. (7) and (11)). We find that when the policy is trained on data where shorter preference are preferred ($\rho < .5$), the reward model that best matches the policy often has the opposite bias ($b$ is high), and vice versa. Thus, the success of e-DPO may be explained by its ability to distill from reward models that do not suffer from the bias in the policy training data, which is particularly helpful when $\rho \leq .5$ as this bias is also not shared by the oracle RM. We provide the full distribution over reward models for all $\rho$ and $\beta$ in Appendix B.3. Overall, these results demonstrate the efficacy of training a policy by distilling from a reward model in the presence of distribution shifts, and that a careful design of an ensemble to mitigate spurious correlations can lead to further performance gains.[7]

## 6.3 Additional results in an unbiased setting

To test the ability of our method to perform well on preference tasks where no bias is present, we next run experiments on the Anthropic Helpfulness dataset (Bai et al., 2022). We use a Gemini 1.0 Ultra (Gemini Team, 2024) LLM-as-a-judge model for evaluating win-rates of the policies over both the SFT starting point and the best DPO baseline. As shown in Table 1, in this unbiased setting our distillation objectives can also provide modest gains. Concretely, e-DPO's win rate against the SFT policy is 65.8%, while DPO's win rate is 64.2%. Moreover, comparing e-DPO and DPO directly, e-DPO wins in 49.7% of the cases, while DPO wins in 46.9% of the cases (the rest are considered to be ties with no preference relation).

> **Takeaways: Experimental results**
>
> Reward model distillation, specifically pessimistic reward model distillation when an ensemble of reward models is used, leads to improved robustness on tasks where bias is present in the preference data. In addition, reward model distillation results in small improvements in performance even on unbiased settings, making it a simple but compelling algorithmic modification to offline training.

## 7 Theoretical analysis

This section characterizes solutions offered by pessimistic DPO and distillation to the issues identified in §4, focusing on the simplified scenario in which we optimize with respect to a single preference pair $(y^w, y^\ell)$.

---

[7]We note that we also experimented with an ensemble where members are different checkpoints across training of a reward model on the preference data, however, we did not observe any empirical gains from this form of ensemble.

| Method | %Wins | %Ties | %Losses |
|---|---|---|---|
| e-DPO vs. SFT | **65.8** | 1.3 | 32.9 |
| d-DPO vs. SFT | 65.6 | 1.0 | 33.3 |
| DPO vs. SFT | 64.2 | 1.0 | 34.8 |

| Method | %Wins | %Ties | %Losses |
|---|---|---|---|
| e-DPO vs. DPO | **49.7** | 3.4 | 46.9 |
| d-DPO vs. DPO | 48.2 | 3.6 | 48.2 |

Table 1: Side-by-side win rates on the Helpfulness dataset (with a Gemini 1.0 Ultra evaluator).

### 7.1 Optima

In its Lagrangian formulation, pessimistic DPO adds a forward KL term to the DPO objective (§5.3). Here we seek to better analyze how this additional term effects the optimal policy. For the sake of analysis, we assume that the preference annotations are sampled from the reference distribution, $\mu(x) \times \pi_{\text{ref}}(y \mid x) \times \pi_{\text{ref}}(y \mid x)$. Then a finite-sample approximation of the forward KL term is

$$\hat{\Omega}(\Theta) := \sum_{(y^w, y^\ell) \in \mathcal{D}_{\text{Pref}}} -(\log \pi_\theta(y^\ell) + \log \pi_\theta(y^w)).$$

By applying this finite-sample approximation, *p-DPO has a finite optimum, unlike DPO*, as shown in Proposition 1. Note that this analysis is limited in two ways: (1) as mentioned, we compute the KL term over the completions in the preference data; (2) we directly optimize the probability ratios $\psi_w = \pi_\theta(y^w)/\pi_{\text{ref}}(y^w)$ and $\psi_\ell = \pi_\theta(y^\ell)/\pi_{\text{ref}}(y^\ell)$, rather than optimizing them jointly through the parameters. For sufficiently expressive $\pi_\theta$, however, this approximation captures the behavior of the two algorithms reasonably well.

**Proposition 5.** *Let $\hat{\mathcal{L}}_{\text{pdpo}}$ represent a finite-sample approximation to $\mathcal{L}_{\text{pdpo}}$ with the empirical forward KL term $\hat{\Omega}(\Theta)$. For a fixed $\hat{\pi}_\theta(y_i^w)$ and $\alpha > 1$, the $\arg\min_{\pi_\theta(y^\ell)} \hat{\mathcal{L}}_{\text{pdpo}}$ is $\min\left(1 - \hat{\pi}_\theta(y_i^w), \hat{\pi}_\theta(y_i^\ell)\right)$, with $\log \hat{\pi}_\theta(y_i^\ell) = -\frac{1}{\beta} \log(\alpha - 1) + \log \hat{\pi}_\theta(y_i^w) + \log \frac{\pi_{\text{ref}}(y_i^\ell)}{\pi_{\text{ref}}(y_i^w)}$.*

The optimum in Proposition 5 corresponds to $\log \psi_w/\psi_\ell = \beta^{-1}\log(\alpha-1)$. Recall that IPO (Azar et al., 2024) seeks to assign a constant value to this ratio by minimizing $(\log \frac{\psi_w}{\psi_\ell} - \tau^{-1})^2$; the (unconstrained) optima are identical for $\tau^{-1} := \beta^{-1}\log(\alpha-1)$, but the loss surfaces are different (see further analysis of this in §7.2). DPO sets $\pi_\theta(y_i^\ell) \to 0$, as shown in Corollary 1; this is due not only to competition from $\pi_\theta(y_i^w)$ but from DPO penalizing positive probability on $y_i^\ell$. Analysis of the distilled loss gives a similar result:

**Proposition 6.** *For any fixed $\hat{\pi}_\theta(y_i^w)$ and $\beta > 0$, the $\arg\min$ of the distilled DPO objective in (7) is $\min(1 - \hat{\pi}_\theta(y_i^w), \hat{\pi}_\theta(y_i^\ell))$, with $\log \hat{\pi}_\theta(y_i^\ell) = \frac{1}{\beta}(r_t(x, y_i^\ell) - r_t(x, y_i^w)) + \log \hat{\pi}_\theta(y_i^w) + \log \frac{\pi_{\text{ref}}(y_i^\ell)}{\pi_{\text{ref}}(y_i^w)}$.*

While the setting is simplistic, the results are comforting: here the additional regularization effects of both distillation and pessimism (in the case of p-DPO) clearly help to avoid degenerate optima.

### 7.2 Transitive closure: p-DPO vs IPO

As pointed out in §7.1, both p-DPO and IPO target a constant ratio for $\log \psi_w/\psi_l$. Despite the similar form of optima, however, the loss surfaces of the two objectives differ in notable ways. To see this, we consider a simplified setting with three possible outputs, $y_1, y_2, y_3$. We observe either $\mathcal{D} = \{(y_1 \prec y_2), (y_2 \prec y_3)\}$ or $\overline{\mathcal{D}} = \mathcal{D} \cup \{(y_1 \prec y_3)\}$. If we treat this problem as a multi-arm bandit, the goal is to assign a weight to each arm, which we denote $\psi_i = \log \pi_\theta(y_i \mid x) + Z_x$, with $Z_x$ an underdetermined log-partition function.

**Proposition 7.** *Let $\mathcal{D} = \{(i, i+1) : i \in 1, 2, \ldots, n\}$ for $n > 2$. Let $\overline{\mathcal{D}}$ be the dataset arising from the transitive closure of $\mathcal{D}$. Assume $\pi_{\text{ref}}$ is indifferent to all $(y_i, y_j)$. Let $\psi_\infty^{(\mathcal{D})} = \max_i \psi_i^{(\mathcal{D})} - \min_i \psi_i^{(\mathcal{D})}$. Then $\psi_\infty^{(\mathcal{D})} = (n-1)\tau^{-1} > \psi_\infty^{(\overline{\mathcal{D}})} = 2\frac{n-1}{n}\tau^{-1}$.*

Intuitively, the observation of $y_1 \prec y_3$ should increase confidence that $y_3$ is superior to $y_1$, but in IPO it has the opposite effect, drawing their scores closer together. While pessimistic DPO also has a target ratio between each preference pair, its loss surface is different: in particular, it does not increase quadratically as we move away from the target. We find empirically that pessimistic DPO is robust to the transitive closure

of preference annotations in the multi-arm bandit setting, as shown in Figure B.2. As discussed above, DPO will set $\psi_1 \to -\infty$ because $y_1$ is never preferred. Specifically, in our experiments we solve the p-DPO and IPO objectives for both $\mathcal{D} = \{(y_1, y_2), (y_2, y_3)\}$ and $\overline{\mathcal{D}} = \mathcal{D} \cup \{(y_1, y_3)\}$, solving with respect to $\{\pi_\theta(y_i)\}$. IPO is solved analytically as a quadratic program; for p-DPO we used projected gradient descent. We consider $\beta \in (1, 3, 10, 30)$ and $\alpha \in (5, 10, 20, 50, 100, 1000)$. As shown in Figure B.2, there are significant differences in the IPO solutions with and without transitive closure, while for p-DPO these differences are imperceptible.

> **Takeaways: Theoretical analysis**
>
> Both pessimistic DPO and reward model distillation avoid the degenerate optima of DPO that were analyzed in §4—primarily through adding additional sources of regularization to the objective. In addition to these more stable optima, we also show how the loss surface of p-DPO can lead to more favorable outcomes with respect to modeling challenging transitive preferences vs. IPO.

## 8    Conclusion

LM alignment is crucial for deploying safe and helpful assistants, but is difficult due to lack of access to perfect preference oracles. We presented a thorough theoretical analysis of some of the degeneracies that DPO is susceptible to when learning from sampled human preference data. Furthermore, our findings suggest that explicit reward modeling remains a powerful vehicle for introducing regularization into post-training. By distilling the reward assigned by a single, explicit reward model—or a family of explicit reward models—directly into the implicit reward maximized by our policies using offline data, we demonstrated that we can achieve improved robustness to variations in preference dataset quality, while maintaining the simplicity of offline alignment frameworks. Finally, reward model distillation also results in modest but consistent improvements in performance even on unbiased settings, making it an overall compelling algorithmic modification to offline training.

## Acknowledgements

We thank the anonympous reviewers, Alexander D'Amour, and Chris Dyer for helpful comments and feedback on the manuscript. This research also benefited from discussions with Victor Veitch, Mandar Joshi, Kenton Lee, Kristina Toutanova, David Gaddy, Dheeru Dua, Yuan Zhang, Tianze Shi, and Anastasios Angelopoulos.

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

# A Proofs

## A.1 Proof of Proposition 1

**Proposition** (Proposition 1 restated). *Under Assumption 1, for any $(y, y')$ such that $y = y_i^w$ and $y' = y_i^\ell$ for some $i$, we have $\frac{\pi_{\theta^*}(y)\pi_{\mathrm{ref}}(y')}{\pi_{\theta^*}(y')\pi_{\mathrm{ref}}(y)} \to \infty$, for all global minimizers $\pi_{\theta^*}$ of the DPO objective in* (6), *for any $\beta > 0$.*

*Proof.* Since all the preference pairs $(y, y')$ are mutually disjoint, and $\theta_y$ is specific to each $y$, the DPO objective over $\mathcal{D}_{\mathrm{pref}}$ is convex in $\Delta = \{\Delta_1, \ldots, \Delta_n\}$, where

$$\Delta_i = \beta \log \frac{\pi_\theta(y_i^w)\pi_{\mathrm{ref}}(y_i^\ell)}{\pi_\theta(y_i^\ell)\pi_{\mathrm{ref}}(y_i^w)}. \tag{15}$$

Furthermore, the different $\Delta_i$ are completely independent from each other due to the preference pairs being disjoint, so they can be optimized over separately.

In particular, for every $i$ we have that

$$\lim_{\Delta_i \to \infty} -\log\left(\sigma\left(\Delta_i\right)\right) = 0, \tag{16}$$

which implies that $\Delta^* = \{\infty\}^n$ is the unique global minimizer of the DPO loss over $\mathcal{D}_{\mathrm{pref}}$ in the space of $\Delta$'s, and any $\theta^*$ that is a global minimizer must therefore satisfy

$$\log \frac{\pi_\theta(y_i^w)\pi_{\mathrm{ref}}(y_i^\ell)}{\pi_\theta(y_i^\ell)\pi_{\mathrm{ref}}(y_i^w)} = \infty. \tag{17}$$

$\square$

## A.2 Proof of Corollary 1

**Corollary** (Corollary 1 restated). *Under Assumption 1, further assume that $0 < \pi_{\mathrm{ref}}(y) < 1$ for all $y$. Then $\pi_{\theta^*}$ is a global minimizer of the DPO objective in* (6) *iff $\pi_{\theta^*}(\mathcal{C}(y^\ell)^c) \to 1$ with $\pi_{\theta^*}(y_i^w) > 0 \ \forall i \in [n]$, where $\mathcal{C}(y^\ell)^c$ is the complement of the set of all responses $y$ that appear as a dispreferred $y_i^\ell$ for any $i \in [n]$.*

*Proof.* Following the same argument of the proof of Proposition 1, we have that all global minimizers $\theta^*$ of the DPO satisfy $\Delta_i^* = \infty$, which in turn implies that

$$\frac{\pi_{\theta^*}(y_i^w)\pi_{\mathrm{ref}}(y_i^\ell)}{\pi_{\theta^*}(y_i^\ell)\pi_{\mathrm{ref}}(y_i^w)} = \infty. \tag{18}$$

Since $\pi_{\mathrm{ref}}(y)$ is assumed to satisfy $0 < \pi_{\mathrm{ref}}(y) < 1$ for all $y$, this implies that all $\theta^*$ satisfy

$$\frac{\pi_{\theta^*}(y_i^w)}{\pi_{\theta^*}(y_i^\ell)} = \infty, \tag{19}$$

which further implies that $\pi_{\theta^*}(y_i^\ell) = 0$ and $\pi_{\theta^*}(y_i^w) > 0$ for all $i \in [n]$, as $\pi_{\theta^*}(y_i^w) \leq 1$ for any $y_i^w$. Aggregating

$$\mathcal{C}(y_\ell) = \{y \colon \exists i \in [n] \text{ s.t } y_i^\ell = y\} \tag{20}$$

then gives that

$$\pi_{\theta^*}(\mathcal{C}(y_\ell)) = \sum_{y \in \mathcal{C}(y_\ell)} \pi_{\theta^*}(y) = 0 \implies \pi_{\theta^*}(\mathcal{C}(y_\ell)^c) = 1. \tag{21}$$

$\square$

To prove the converse, let $\pi_{\theta'}$ be a policy that satisfies $\pi_{\theta'}(\mathcal{C}(y^\ell)^c) = 1$, with $\pi_{\theta'}(y_i^w) > 0$, $\forall i \in [n]$,. As $\pi_{\theta'}(y) \geq 0$ for all $y$, this implies that $\pi_{\theta'(y_i^\ell)} = 0$ $\forall i \in [n]$. Then, we have

$$\frac{\pi_{\theta'}(y_i^w)}{\pi_{\theta'}(y_i^\ell)} = \infty, \tag{22}$$

which by Proposition 1 implies that $\pi_{\theta'}$ is a global optimum.

## A.3 Proof of Theorem 1

**Theorem** (Theorem 1 restated). *Let $\mathcal{Y}$ denote the set of all possible responses for any model $\pi_\theta$. Assume that $\text{supp}(\pi_{\text{ref}}(y \mid x)) = \mathcal{Y}$, i.e., the reference policy may generate any outcome with non-zero probability. Further, let $\text{supp}(\rho(x, y_1, y_2)) = \text{supp}(\mu(x)) \times \mathcal{Y} \times \mathcal{Y}$. Let $\pi_{\theta*}(y \mid x) \in \arg\min_{\pi_\theta} \mathcal{L}_{\text{distill}}(r^*, \pi_\theta; \rho)$ be a minimizer over all possible policies, of the implicit reward distillation loss in (7), for which $r^*(x, y)$ is assumed to be deterministic, and finite everywhere. Then for any $\beta > 0$, $\pi_{\theta*}$ also maximizes the alignment objective in (1).*

*Proof.* We know that the optimal policy for the RLHF objective (1) is given by $\pi_{\theta*}(y|x) \propto \pi_{\text{ref}}(y|x) \exp(r^*(x, y)/\beta)$. Plugging this policy into the distillation objective (7), we see that $\mathcal{L}_{\text{distill}}(r^*, \pi_{\theta*}, \rho) = 0$ for all $\rho$. In fact, the loss is equal to 0 pointwise, meaning that $\pi_{\theta*}$ is a global minimizer of the distillation objective (7). Further, let $\pi$ be some other minimizer of $\mathcal{L}_{\text{distill}}(r^*, \cdot, \rho)$. Then $\pi$ also has to attain a loss of 0 at all $(x, y, y')$ in the support of $\rho$, meaning that $\log \pi(y|x) - \log \pi(y'|x) = \log \pi_{\theta*}(y|x) - \log \pi_{\theta*}(y|x)$ for all $(x, y, y')$ in the support of $\rho$. Consequently, the two policies coincide in the support of $\rho$ (due to the normalization constraint, there is no additional offset term allowed as the support of $\rho$ covers all of $\mathcal{Y}$). Finally, noting that the support of the chosen $\rho$ is such that $\pi_{\theta*}$ puts no mass outside its support due to the KL constraint in (1), we complete the proof. $\square$

## A.4 Proof of Theorem 2

**Theorem** (Theorem 2 restated). *Define the constrained minimizer*

$$\pi_{\theta*}(y \mid x) \in \underset{\pi_\theta \in \mathcal{P}_\beta(\mathcal{S})}{\arg\min} \beta \mathbb{E}_{\mu(x)} \mathbb{D}_{\text{KL}}(\pi_{\text{ref}}(\cdot \mid x) \| \pi_\theta(\cdot \mid x)),$$

*where $\mathcal{P}_\beta(\mathcal{S})$ is the set of all possible policies with implicit reward models that are consistent with any target reward model $r_{\text{tgt}}^i \in \mathcal{S}$, i.e., $\mathcal{P}_\beta(\mathcal{S}) \triangleq \{\pi_{\theta_i}\}_{i=1}^{|\mathcal{S}|}$ where $\pi_{\theta_i} \propto \pi_{\text{ref}}(y \mid x) \exp \frac{1}{\beta} r_{\text{tgt}}^i(x, y)$. Then for any $\beta > 0$, $\pi_{\theta*}$ also maximizes the pessimistic alignment objective in (8).*

*Proof.* Consider the pessimistic objective:

$$\max_{\pi_\theta} \min_{r_{\text{tgt}} \in \mathcal{S}} \mathbb{E}_{\mu(x)} \Big[ \mathbb{E}_{\pi_\theta(y|x)}[r_{\text{tgt}}(x, y)] - \mathbb{E}_{\pi_{\text{ref}}(y|x)}[r_{\text{tgt}}(x, y)] \Big] - \beta \mathbb{D}_{\text{KL}}(\pi_\theta \| \pi_{\text{ref}}). \tag{23}$$

As it is linear in $r_{\text{tgt}}$ and convex in $\pi$, we can switch the order of min and max:

$$\min_{r_{\text{tgt}} \in \mathcal{S}} \left[ \max_{\pi \in \Pi} \mathbb{E}_{\mu(x)} \Big[ \mathbb{E}_{\pi(y|x)}[r_{\text{tgt}}(x, y)] - \mathbb{E}_{\pi_{\text{ref}}(y|x)}[r_{\text{tgt}}(x, y)] \Big] - \beta \mathbb{D}_{\text{KL}}(\pi \| \pi_{\text{ref}}) \right]. \tag{24}$$

Note that every $r_{\text{tgt}} \in \mathcal{S}$ can be written in terms of the KL-constrained policy $\pi_{r_{\text{tgt}}}^*$ it induces, i.e.,

$$r_{\text{tgt}}(x, y) = \beta \log \frac{\pi_{r_{\text{tgt}}}^*(y \mid x)}{\pi_{\text{ref}}(y \mid x)} + \beta \log Z(x, r_{\text{tgt}}), \tag{25}$$

where

$$\pi_{r_{\text{tgt}}}^* = \underset{\pi_\theta}{\arg\max} \, \mathbb{E}_{\mu(x)} \mathbb{E}_{\pi_\theta(y|x)}[r_{\text{tgt}}(x, y)] - \beta \mathbb{D}_{\text{KL}}(\pi_\theta \| \pi_{\text{ref}}) \tag{26}$$

which has the form

$$\pi^*_{r_{\text{tgt}}}(y \mid x) = \frac{1}{Z(x, r_{\text{tgt}})} \pi_{\text{ref}}(y \mid x) \exp\left(\frac{1}{\beta} r_{\text{tgt}}(x, y)\right) \tag{27}$$

where $Z(x, r_{\text{tgt}})$ is the partition function:

$$Z(x, r_{\text{tgt}}) = \sum_{y \in \mathcal{Y}} \pi_{\text{ref}}(y \mid x) \exp\left(\frac{1}{\beta} r_{\text{tgt}}(x, y)\right). \tag{28}$$

Substituting $\pi^*_{r_{\text{tgt}}}$ in for $\max_\pi$ and writing $r_{\text{tgt}}$ in terms of $\pi^*_{r_{\text{tgt}}}$, we get the simplified objective

$$
\begin{aligned}
\min_{r_{\text{tgt}} \in \mathcal{S}} & \left[ \max_{\pi \in \Pi} \mathbb{E}_{\mu(x)} \Big[ \mathbb{E}_{\pi(y|x)}[r_{\text{tgt}}(x, y)] - \mathbb{E}_{\pi_{\text{ref}}(y|x)}[r_{\text{tgt}}(x, y)] \Big] - \beta \mathbb{D}_{\text{KL}}(\pi \| \pi_{\text{ref}}) \right] \\
&= \min_{r_{\text{tgt}} \in \mathcal{S}} \left[ \mathbb{E}_{\mu(x)} \left[ \mathbb{E}_{\pi^*_{r_{\text{tgt}}}(y|x)} \left[ \beta \log \frac{\pi^*_{r_{\text{tgt}}}(y \mid x)}{\pi_{\text{ref}}(y \mid x)} + \beta \log Z(x, r_{\text{tgt}}) \right] \right.\right. \\
& \qquad\qquad\qquad - \mathbb{E}_{\pi_{\text{ref}}(y|x)} \left[ \beta \log \frac{\pi^*_{r_{\text{tgt}}}(y \mid x)}{\pi_{\text{ref}}(y \mid x)} + \beta \log Z(x, r_{\text{tgt}}) \right] \\
& \qquad\qquad\qquad \left.\left. - \beta \mathbb{D}_{\text{KL}}(\pi^*_{r_{\text{tgt}}} \| \pi_{\text{ref}} \mid x) \right] \right] \\
&= \min_{r_{\text{tgt}} \in \mathcal{S}} \beta \left[ \mathbb{E}_{\mu(x)} \left[ \mathbb{D}_{\text{KL}}(\pi^*_{r_{\text{tgt}}} \| \pi_{\text{ref}} \mid x) + \mathbb{D}_{\text{KL}}(\pi_{\text{ref}} \| \pi^*_{r_{\text{tgt}}} \mid x) - \mathbb{D}_{\text{KL}}(\pi^*_{r_{\text{tgt}}} \| \pi_{\text{ref}} \mid x) \right] \right] \\
&= \min_{r_{\text{tgt}} \in \mathcal{S}} \beta \mathbb{E}_{\mu(x)} \left[ \mathbb{D}_{\text{KL}}(\pi_{\text{ref}} \| \pi^*_{r_{\text{tgt}}} \mid x) \right].
\end{aligned}
\tag{29}
$$

$\square$

## A.5 Proof of Proposition 2

**Proposition** (Proposition 2 restated). *Let $y^w, y^\ell \in \mathcal{V}^n$ be preferred versus dispreferred outputs of length $n$, respectively, with $\pi_{\text{ref}}(y^w), \pi_{\text{ref}}(y^\ell) > 0$ and corresponding count vectors $c(y^w), c(y^\ell)$. Let $\log \pi_\theta(y) = c(y) \cdot \theta - nZ(\theta)$ for $Z(\theta) = \log \sum_i^\mathcal{V} e^{\theta_i}$, with upper bound $\log \tilde{\pi}_\theta(y) = c(y) \cdot \theta - n \max_j \theta_j$. Let $\theta^{(t)}$ represent the parameters of $\pi$ after $t$ steps of gradient descent on $\mathcal{L}_{\text{dpo}}(\{y^\ell, y^w, x\})$, with $\theta^{(0)} = 0$. Then, we have that*

$$\pi_{\theta^{(t)}}(y^w) \leq \tilde{\pi}_{\theta^{(t)}}(y^w) \leq \tilde{\pi}_{\theta^{(t-1)}}(y^w) \quad \text{for all } t,$$

*with strict inequality when $\|c(y^w) - c(y^\ell)\|_0 > 1$.*

*Proof.* Let $\Delta = [c(y^w) - c(y^\ell)]$ and $\rho = \pi_{\text{ref}}(y^w)/\pi_{\text{ref}}(y^\ell)$. The theorem assumes $|y^w| = |y^\ell|$. Then $\mathcal{L}_{\text{dpo}} = -\log \sigma(\beta(\Delta \cdot \theta) + \beta \log \rho)$. The derivative with respect to $\theta$ is,

$$\frac{\partial \mathcal{L}_\beta(\theta)}{\partial \theta} = -(1 - \sigma(\beta(\Delta \cdot \theta) + \beta \log \rho))\beta\Delta = -p(y^\ell \succ y^w; \theta)\beta\Delta \prec 0. \tag{30}$$

Let $\delta_t = \beta p(y^\ell \succ y^w; \theta^{(t)})$. Then,

$$
\begin{aligned}
\tilde{\pi}_{\theta^{(t)}} &= \theta^{(t)} \cdot c(y^w) - n \max_j \theta_j^{(t)} \tag{31} \\
&= (\theta^{(t-1)} + \delta_t \Delta) \cdot c(y^w) - n \max_j (\theta_j^{(t-1)} + \delta_t \Delta_j) \tag{32} \\
&= \theta^{(t-1)} \cdot c(y^w) - n \max_j \theta_j^{(t-1)} + \delta_t \Delta \cdot c(y^w) - n\delta_t \max_j \Delta_j \tag{33} \\
&= \tilde{\pi}_{\theta^{(t-1)}} + \delta_t \left( \Delta \cdot c(y^w) - n \max_j \Delta_j \right) \tag{34} \\
&= \tilde{\pi}_{\theta^{(t-1)}} + \delta_t \sum_j^\mathcal{V} c_j(y^w)(\Delta_j - \max_{j'} \Delta_{j'}) \leq \tilde{\pi}_{\theta^{(t-1)}}. \tag{35}
\end{aligned}
$$

We obtain $\max_j \left( \theta_j^{(t-1)} + \delta_t \Delta_j \right) = \max_j \theta_j^{(t-1)} + \max_j \delta_t \Delta_j$ from the fact that $\theta^{(0)} = 0$ and therefore $j \in \arg\max \Delta$ implies $j \in \arg\max \theta^{(t')}$ for all $t' > 0$. The second-to-last step uses $n = \sum_j^{\mathcal{V}} c_j(y^w)$ and the final step uses $\Delta_j \leq \max_{j'} \Delta_{j'}$. Finally, we have $\pi_{\theta^{(t)}}(y) \leq \tilde{\pi}_{\theta^{(t)}}(y^w)$ because $Z(\theta) = \log \sum_j \exp \theta_j \geq \log \max_j \exp \theta_j = \max_j \theta_j$. $\qquad\square$

## A.6 Proof of Proposition 3

**Proposition** (Proposition 3 restated). *Let $y^w$ and $y^\ell$ be preferred versus dispreferred outputs of length $n$. Let $\Delta = c(y^w) - c(y^\ell)$ be the difference in unigram counts. Let $\hat{y} = [i, i, \ldots, i]$, for $i \in \arg\max \Delta$, with $\|c(\hat{y})\|_1 = n$. Then $\pi_{\theta^{(t)}}(y^w) - \pi_{\theta^{(t)}}(\hat{y}) = \tau(t)k$ for some $k \leq 0$ and some non-decreasing $\tau : \mathbb{Z}_+ \to \mathbb{R}_+$.*

*Proof.* Applying gradient descent with learning rate $\eta$ to the gradient from Equation (30), at each step $t$ the parameters are,

$$\theta^{(t)} = \theta^{(t-1)} + \eta\beta p(y^\ell \succ y^w; \theta^{(t-1)})\Delta = \left( \sum_{t'=1}^{t} \eta\beta p(y^\ell \succ y^w; \theta^{(t')}) \right)\Delta = \tau(t)\Delta. \tag{36}$$

Plugging these parameters into the likelihoods,

$$\ell_{\theta^{(t)}}(c(y^w)) - \ell_{\theta^{(t)}}(\hat{y}) = c(y^w) \cdot \theta^{(t)} - nZ(\theta^{(t)}) - c(\hat{y}) \cdot \theta^{(t)} + nZ(\theta^{(t)}) \tag{37}$$

$$= (c(y^w) - c(\hat{y})) \cdot \theta^{(t)} = (c(y^w) - c(\hat{y})) \cdot (\tau(t)\Delta) \tag{38}$$

$$= \tau(t)(c(y^w) \cdot \Delta - n \max \Delta) = \tau(t)k, \tag{39}$$

with $k \leq 0$ by $c(y^w) \cdot \Delta \leq \|c(y^w)\|_1 \times \|\Delta\|_\infty = n \max \Delta$. $\qquad\square$

## A.7 Proof of Proposition 4

**Proposition** (Proposition 4 restated). *Assume the same conditions as Theorem 1. Then for any $0 < \gamma < \infty$, there exists a $\lambda \geq 0$ such that $\pi_{\theta^*}(y \mid x) \in \arg\min_{\pi_\theta} \mathcal{L}_{\mathrm{pdistill}}(\mathcal{S}, \pi_\theta; \rho)$, where $\pi_{\theta^*}$ is a minimizer over all possible policies of the objective* (9), *for the effective reward model set*

$$\mathcal{S}_\gamma = \bigcup_{r_{\mathrm{tgt}}^i \in \mathcal{S}} \left\{ \tilde{r} \colon \mathbb{E}_{\rho(x, y_1, y_2)} \left[ (r_{\mathrm{tgt}}^i(x, y_1) - r_{\mathrm{tgt}}^i(x, y_2) - \tilde{r}(x, y_1) + \tilde{r}(x, y_2))^2 \right] \leq \lambda \right\}.$$

*Proof.* The proof is a standard Lagrangian duality argument, which we reproduce here for completeness. For two functions $f(z)$ and $g(z)$, let us define

$$z^* = \arg\min_z f(z) + \alpha g(z). \tag{40}$$

Let us also consider the constrained problem

$$z' = \arg\min_z f(z) \quad \text{s.t. } g(z) \leq g(z^*). \tag{41}$$

Suppose by contradiction that $z^*$ is not a minimizer of (41). Since $z^*$ is feasible for the constraint by construction, we get that $f(z') < f(z^*)$. Consequently, we further have

$$f(z') + \alpha g(z') < f(z^*) + \alpha g(z^*),$$

where the inequality follows from the feasibility of $z'$ in (41). This contradicts the optimality of $z^*$ in (40), meaning that $z^*$ must be a minimizer of (41). Applying this general result with $f = \beta\mathbb{E}_{\mu(x)}\mathbb{D}_{\mathrm{KL}}(\pi_{\mathrm{ref}}(y \mid x)\|\pi_\theta(y \mid x))$, $g = \min_{r_{\mathrm{tgt}}^i \in \mathcal{S}} \mathcal{L}_{\mathrm{distill}}(r_{\mathrm{tgt}}^i, \pi_\theta; \rho)$, and $z = \pi_\theta$ completes the proof, since we recognize the set $\mathcal{S}_\gamma$ in (12) to be equivalent to $\bigcup_{r_{\mathrm{tgt}}^i \in \mathcal{S}} \mathcal{L}_{\mathrm{distill}}(r_{\mathrm{tgt}}^i, \pi_\theta; \rho) \leq \lambda$. $\qquad\square$

### A.8 Proof of Proposition 5

**Proposition** (Proposition 5 restated). *Let $\hat{\mathcal{L}}_{\mathrm{pdpo}}$ represent a finite-sample approximation to $\mathcal{L}_{\mathrm{pdpo}}$ with the empirical forward KL term $\hat{\Omega}(\Theta)$. For a fixed $\hat{\pi}_\theta(y_i^w)$ and $\alpha > 1$, the $\operatorname{argmin}_{\pi_\theta(y^\ell)} \hat{\mathcal{L}}_{\mathrm{pdpo}}$ is* $\min\left(1 - \hat{\pi}_\theta(y_i^w), \hat{\pi}_\theta(y_i^\ell)\right)$, *with* $\log \hat{\pi}_\theta(y_i^\ell) = -\frac{1}{\beta}\log(\alpha - 1) + \log \hat{\pi}_\theta(y_i^w) + \log \frac{\pi_{\mathrm{ref}}(y_i^\ell)}{\pi_{\mathrm{ref}}(y_i^w)}$.

*Proof.* We differentiate $\mathcal{L}_{\mathrm{pdpo}}$ with respect to $\psi_\ell = \pi_\theta(y^\ell)/\pi_{\mathrm{ref}}(y^\ell)$ with $i$ implicit, obtaining,

$$\frac{\partial \mathcal{L}_{\mathrm{pdpo}}}{\partial \psi_\ell} = \beta \frac{\psi_\ell^\beta}{\psi_w^\beta + \psi_\ell^\beta}\psi_\ell^{-1} - \frac{\beta}{\alpha}\psi_\ell^{-1} = \beta\psi_\ell^{-1}\left(\frac{\psi_\ell^\beta}{\psi_w^\beta + \psi_\ell^\beta} - \alpha^{-1}\right) \tag{42}$$

which is zero when,

$$\alpha\psi_\ell^\beta = \psi_w^\beta + \psi_\ell^\beta \tag{43}$$

$$\psi_\ell = \left(\frac{1}{\alpha - 1}\right)^{1/\beta}\psi_w \tag{44}$$

$$\log \psi_\ell = -\frac{1}{\beta}\log(\alpha - 1) + \log \psi_w \tag{45}$$

$$\log \pi_{\hat{\theta}}(y^\ell) = \log \pi_{\mathrm{ref}}(y^\ell) - \frac{1}{\beta}\log(\alpha - 1) + \log \pi_\theta(y^w) - \log \pi_{\mathrm{ref}}(y^w). \tag{46}$$

By the second-order condition, the critical point is a minimum. The objective $\mathcal{L}_{\mathrm{pdpo}}$ is the sum of two components: the negative log sigmoid term for $\mathcal{L}_i$ and the negative log probability for $\hat{\Omega}$. Because each component is a convex function of $\psi_i$, so is $\mathcal{L}_{\mathrm{pdpo}}$. As a result, the local minimum $\log \hat{\pi}_\theta(y^\ell)$ is also a global minimum. □

### A.9 Proof of Proposition 6

**Proposition** (Proposition 6 restated). *For any fixed $\hat{\pi}_\theta(y_i^w)$ and $\beta > 0$, the $\operatorname{argmin}$ of the distilled DPO objective in (7) is* $\min(1 - \hat{\pi}_\theta(y_i^w), \hat{\pi}_\theta(y_i^\ell))$, *with* $\log \hat{\pi}_\theta(y_i^\ell) = \frac{1}{\beta}(r_t(x, y_i^\ell) - r_t(x, y_i^w)) + \log \hat{\pi}_\theta(y_i^w) + \log \frac{\pi_{\mathrm{ref}}(y_i^\ell)}{\pi_{\mathrm{ref}}(y_i^w)}$.

*Proof.* This follows directly from differentiating (7) with respect to $\pi_\theta(y_2)$. □

### A.10 Proof of Proposition 7

**Proposition** (Proposition 7 restated). *Let $\mathcal{D} = \{(i, i+1) : i \in 1, 2, \ldots, n\}$ for $n > 2$. Let $\overline{\mathcal{D}}$ be the dataset arising from the transitive closure of $\mathcal{D}$. Assume $\pi_{\mathrm{ref}}$ is indifferent to all $(y_i, y_j)$. Let $\psi_\infty^{(\mathcal{D})} = \max_i \psi_i^{(\mathcal{D})} - \min_i \psi_i^{(\mathcal{D})}$. Then $\psi_\infty^{(\mathcal{D})} = (n-1)\tau^{-1} > \psi_\infty^{(\overline{\mathcal{D}})} = 2\frac{n-1}{n}\tau^{-1}$.*

*Proof.* For $\mathcal{D}$, the IPO objective can be minimized at zero, so that $\psi_\infty^{(\mathcal{D})} = (n-1)\tau^{-1}$. For $\overline{\mathcal{D}}$, each adjacent pair of completions is separated by $\gamma$, and the objective is $\sum_{i=1}^{n-1}(n-i)(i\gamma - \tau^{-1})^2$. The minimum is $\gamma = \frac{n(n+1)(n-1)/6}{n^2(n+1)(n-1)/12}\tau^{-1} = \frac{2}{n}\tau^{-1}$, so that $\psi_\infty^{(\overline{\mathcal{D}})} = (n-1)\gamma = 2\frac{n-1}{n}\tau^{-1} < (n-1)\tau^{-1} = \psi_\infty^{(\mathcal{D})}$ for $n > 2$. □

## B Additional results

### B.1 The effect of distillation and pessimism on likelihood collapse

Figure B.1 tests the effect of pessimism in both DPO (DPO vs p-DPO) and distilled DPO (d-DPO vs dp-DPO) on likelihood collapse in the preferred vs. dispreferred generations in the offline data. Note that a subtle point arises from the distinction between regularizing to the reference policy output distribution versus regularizing to the preference data distribution, which are only (asymptotically) identical if preferences are annotated on

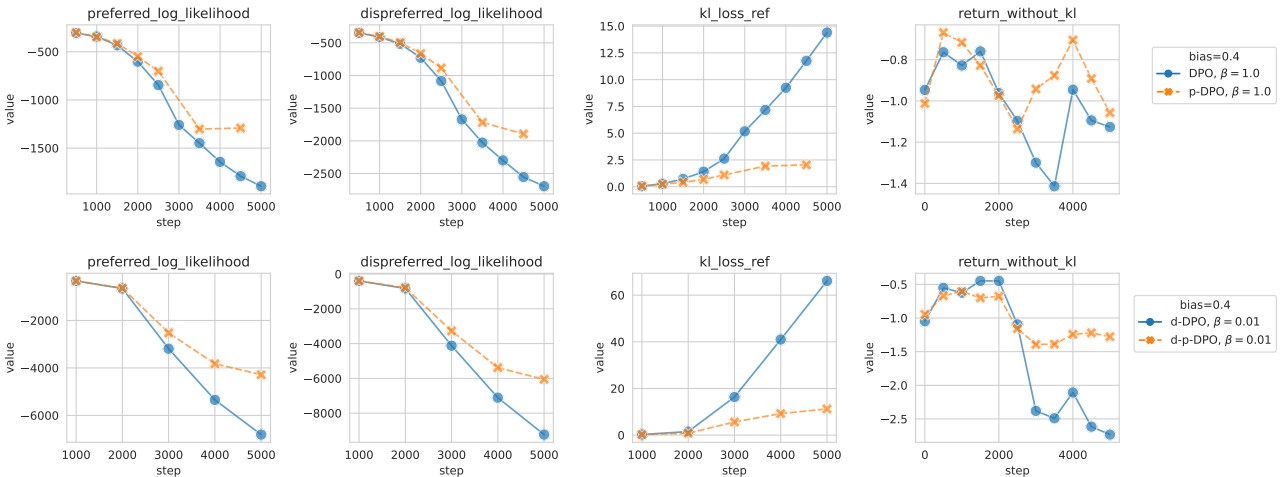

Figure B.1: Pessimism mitigates likelihood collapse. In this case, we penalize the *sample* KL-divergence rather than the *data* KL-divergence, because the samples are not drawn from the preference data distribution. Thus the effect of regularization is primarily on kl_loss_ref, which quantifies this divergence, rather than on the likelihood of the preference data itself, (dis)preferred_log_likelihood.

a sample from the reference policy (in our initial experiments we had found slightly better results overall when regularizing to the reference policy output distribution). Figure B.1 reports results with respect to both distributions, showing that pessimism (a) mitigates the decrease in probability of preferred and dispreferred preference annotations, despite this data not being used in the regularizers (left-most subplots), and (b) mitigates the increase in KL divergence with respect to the reference distribution (third subplot from left), as expected due to the additional regularization term. As argued in §4 (see also Azar et al. (2024)), the $\beta$ hyperparameter of DPO does not effectively regularize this KL distribution because the implicit DPO reward model assigns infinite-magnitude rewards.

## B.2 Transitive closure

Figure B.2 shows the results of our multi-arm bandit experiments with p-DPO vs. IPO losses, as described in §7.2. In this synthetic setup, we solve the p-DPO and IPO objectives for both $\mathcal{D} = \{(y_1, y_2), (y_2, y_3)\}$ and $\overline{\mathcal{D}} = \mathcal{D} \cup \{(y_1, y_3)\}$, solving with respect to $\{\pi_\theta(y_i)\}$.

## B.3 Distribution over reward models for e-DPO

Figure B.3 investigates the reason for the success of e-DPO, especially when $\rho < .5$. For every length bias, we track during training for all training examples which reward model, $r_{\rho,b}$, best matched the implicit reward of the currently trained e-DPO policy, and plot the distribution over reward models. The policy matches different reward models in different examples. Moreover, there is inverse correlation between the data bias for policy training ($\rho$) and the data bias for training the reward models ($b$). This suggests that the ensemble in e-DPO helps as the policy is distilling from reward models that do not share the data bias of the policy training set.

## B.4 Hyperparameters

Validation set performance across the range of hyperparameter settings is shown in Figure B.4. Figure B.5 also explores the impact of $\gamma$ (which is minimal). In pilot studies we found that these results were relatively robust to variation in the random seed, but did not conduct extensive investigation of this effect across all methods and hyperparameters due to cost.

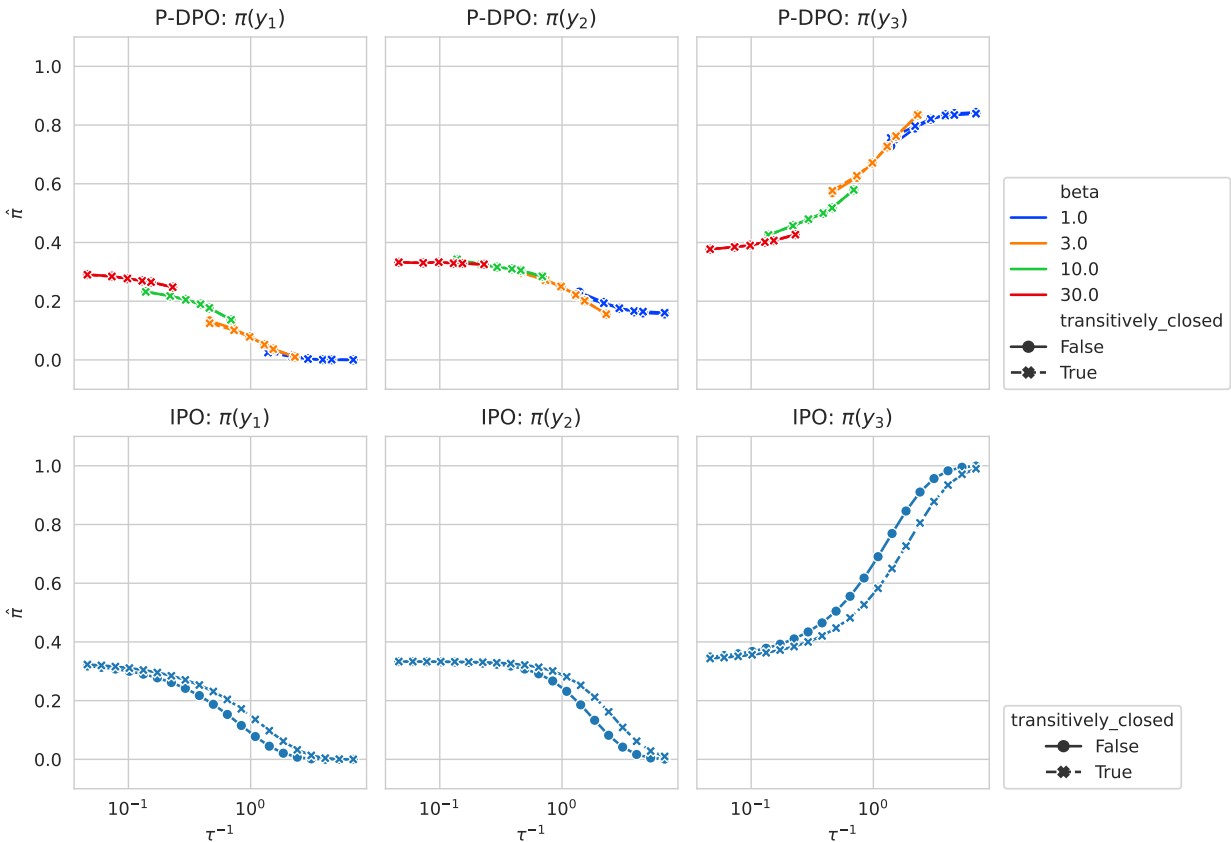

Figure B.2: **Effect of transitive closure on p-DPO and IPO solutions to preference learning in a multi-arm bandit**. Each column shows the learned policy probability for a given arm, based on the preferences $y_1 \prec y_2 \prec y_3$. The top row shows that in p-DPO, the probabilities are not materially affected by the transitive closure $y_1 \prec y_3$. The bottom row shows that in IPO, transitive closure causes the probabilities to be compressed. In each subfigure, we sweep a range of effective values of $\tau^{-1}$, shown on the x-axis.

## C  Compute resources

We train policies on 32 TPU v3 chips and reward models on 16 TPU v3 chips. We obtain roughly 0.1 steps per second when training, for both the policy and reward models.

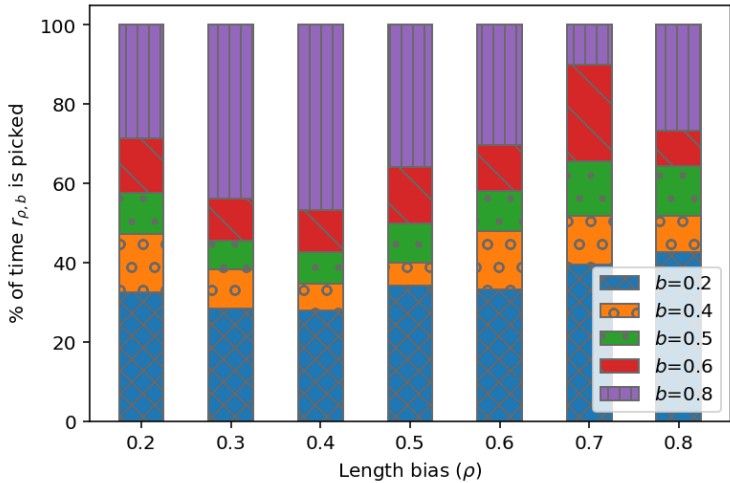

Figure B.3: For all training examples, we track which reward model during training best matches the implicit reward of the current e-DPO policy and plot the distribution over reward models, for every length bias, $\rho$. We observe that the e-DPO policy matches different reward models across examples during training. Moreover, when the policy is trained with data biased towards preferring short responses, the reward model that was trained on longer responses is by and large preferred and vice versa.

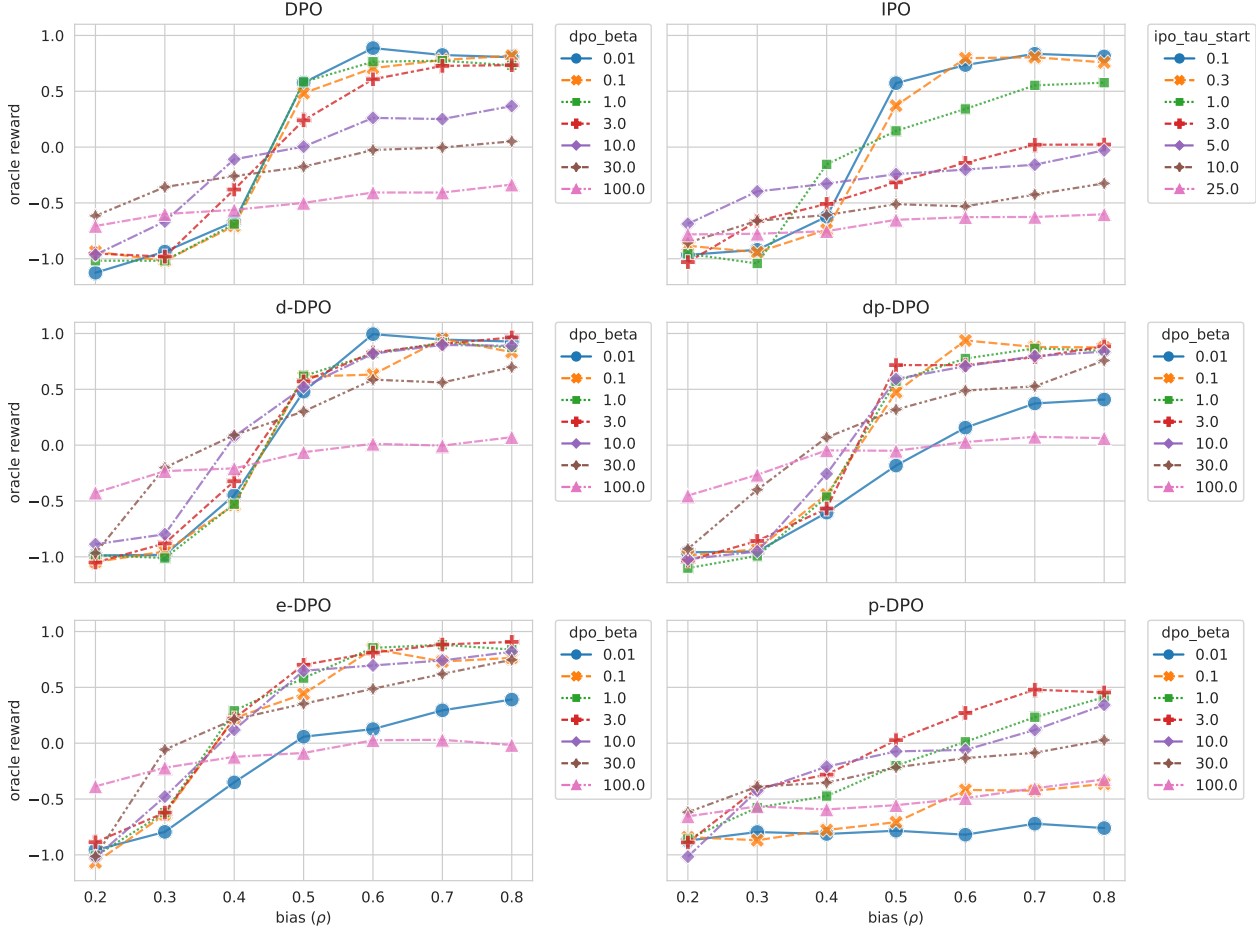

Figure B.4: **Validation set results** across hyperparameters for each method. For all methods, different values of $\rho$ induce different optimal hyperparameters $\beta$ and $\tau^{-1}$.

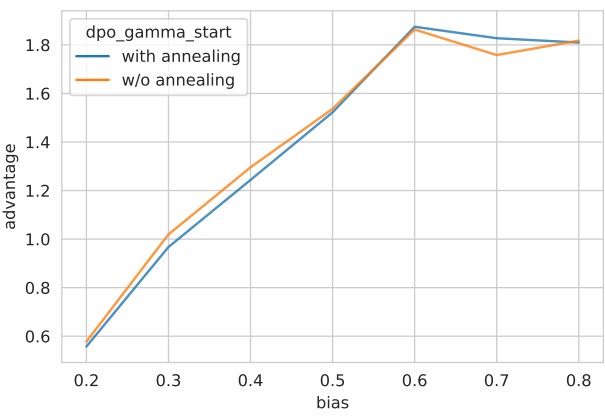

Figure B.5: Comparing e-dpo with and without annealing of $\gamma$ on the development set for all length biases using the best value of $\beta$ per bias. Annealing $\gamma$ has minimal effect on performance and is not necessary for the success of e-dpo.

