# OpenReview forum: "Robust Preference Optimization through Reward Model Distillation"
_TMLR — Accepted by TMLR_

### Review · Reviewer_r5NJ · 2024-11-08

**Summary Of Contributions:**

The paper introduces a robust method of preference optimization that combines the simplicity of direct preference optimization with the benefits of explicit reward modelling. Key contributions include: (1) propose a reward model distillation approach to efficiently train language model policies offline, (2) theoretically and empirically demonstrate improved robustness to distribution shifts in preference data, and (3) introduce a pessimistic reward model distillation that optimizes against a family of reward models to handle uncertainty better.

**Audience:**

Yes

**Claims And Evidence:**

Yes

**Requested Changes:**

1. The description of the reference policy presented in Section 3.1 lacks clarity and could benefit from a more detailed explanation.
2. Could the authors clarify why, under Assumption 1, the ratio $\frac{\pi_{\theta^*}(y) \pi_{ref}(y')}{\pi_{\theta^*}(y') \pi_{ref}(y)}$ tends towards infinity? Additionally, it would be helpful to discuss the prevalence of real-world datasets that meet Assumption 1 and the implications of this for practical applications.
3. The impact of varying the coefficient $\gamma$ should be studied in the experiment section or in the appendix. The manuscript mentions that both annealing and constant $\gamma$ are shown in the experiments with e-DPO. Please refer the section near the text.

**Strengths And Weaknesses:**

This paper starts from rationale and clearly presented motivations. It shows comprehensive experimental setup and analysis that clearly demonstrate the method's efficacy. Finally, it provides strong theoretical backing that addresses and mitigates known issues with existing DPO techniques.

This is a good paper. Some minor weaknesses/questions are shown in the next part.

---

> ### Author Response · Authors · 2024-12-25
>
> We thank the reviewer for all their time spent reviewing this paper, and for their helpful and constructive comments!
>
> > The description of the reference policy presented in Section 3.1 lacks clarity
>
> We have provided a more detailed description in Section 3.1 of the updated draft.
>
> > Could the authors clarify why, under Assumption 1, the ratio tends towards infinity?
>
> Besides the formal proof, intuitively, Assumption 1 states that we have a preference dataset where each response either appears only as a preferred response or a dispreferred response. This would correspond, for example, to a common real-world scenario where for each prompt only two responses A and B are sampled, and given one preference annotation (that is, either A is marked preferred or B is marked preferred). Informally, this results in a degenerate likelihood ratio since the implicit reward, which is defined by the likelihood difference, is trying to match a Bradley Terry model that assigns probability 1 to the preferred sample. Under the DPO reparametrization, this can only be achieved by having an infinite difference in log likelihood ratios.
>
> > On the impact of $\gamma$
>
> We have added the additional result referred to in manuscript (and updated the reference therein) that explores the impact of $\gamma$ to the Appendix. In short, we find that this parameter need not be tuned (and we did not extensively tune it in our experiments). As shown in the new Figure B.5 in the Appendix, a constant gamma of 1e-3 vs. annealing from 1e-4 to 1e-2 leads to very similar results (in fact, even slightly better).

---

> > ### Comment · Reviewer_r5NJ · 2025-01-02
> >
> > Thank you for addressing the questions. All of my concerns have been resolved.

---

### Review · Reviewer_QDvm · 2024-11-15

**Summary Of Contributions:**

The paper studies a shortcoming of  Direct Preference Optimization (DPO) where-in DPO assigns infinite reward value that may lead to degenerate policies. Instead, the paper proposes an approach that updates the model (LM) to match the output distribution of an explicit reward model via Kullback-Liebler regularization term and shows that the regularized methods avoid degenerate solutions. A distillation based approach and a ``pessimistic'' optimization based approach are proposed to train the explicit reward model. The paper empirically compares the new approaches listed above as well as DPO and Identity Preference Optimization (IPO) on datasets that have certain biases as well as on an unbiased dataset. Results suggest that the newly proposed approaches improve performance over IPO and DPO.


The paper observes that

**Audience:**

Yes

**Broader Impact Concerns:**

I do not see a Broader Impact Statement in the paper which is fine.

If the authors feel that they want to share any concerns they are encouraged to do so . This work is very theoretical and as such needs no specific concerns to be noted in my opinion.

**Claims And Evidence:**

Yes

**Requested Changes:**

Following proposed changes, if made, may significantly strengthen the work ("critical")
- Please consider unifying portions of Section 3 and Section 6 as it pertains to analytical work that shows a form of degeneracy in DPO.
- Consider updating caption for Figure 2 to include a small discussion on how a non-specialist can interpret the Y-axis values. ('Why is -0.5 better for \rho = 0.2). I understand that there is a sentence in Page 9 in the draft but the figure may be easier to read if a clarifying sentence or two is included in the caption itself

Following proposed changes, if made, may strengthen the work
- Please consider empirical analysis with additional preference datasets. I am not an expert in post-training research but there appear to be additional datasets that is used to study distribution shifts, for e.g [1]
- Why are other explicit reward model-based approaches not considered for comparison in the paper? A discussion would suffice in the paper.

[1] Lin et al., https://arxiv.org/abs/2409.03650

**Strengths And Weaknesses:**

# Strengths

- The paper analytically studies the problem where-in DPO provides degenerate solutions (policies). The analysis uses a simplified setup to show that DPO exhibits a form of overfitting where-in the preferred output can be assigned a small non-zero probability which can lead to degenerate solutions.

- The paper addresses the shortcoming identified above with a proposed explicit reward model that is used to update the model (LM) by distilling the outputs from the reward model either with single or multiple reward models ("teachers" if I understood this correctly) as well as a pessimistic version of DPO. The key contribution here is the proposed regularization term that seeks to match the distribution of the explicit reward model with that of the model (LM).

- The paper conducts an empirical study to show that the proposed methods (distillation, pessimistic reward model and a combination of the two) show improved performance in cases where it is challenging to learn a good alignment model directly from preference data.

# Weaknesses

- The initial development of showing an analytical case where DPO can provide degenerate solutions is a strength of this paper. However, the development is spread of several seemingly disjointed sections within the main paper itself which may make it unnecessarily harder to follow for a general audience.

- The paper does not make a connection to the issue uncovered via analysis to what is observed in practice even though the empirical shortcoming is mentioned in the paper. Specifically, it would be good to see how the probability mass being moved away from preferred solutions can be resolved with the methods proposed in the  paper. These ablations would be very beneficial to the reader

- The paper studies one dataset that has a bias (length of summary) and shows how some of the newly proposed methods improve over DPO and IPO. Given the use of an explicit reward model, it may make sense to compare with other methods that propose explicit reward models. Or perhaps provide a discussion on why comparing to DPO/IPO is reasonable in the paper.

---

> ### Author Response · Authors · 2024-12-25
>
> We thank the reviewer for all their time spent reviewing this paper, and for their helpful and constructive comments!
>
> Please see our updated revision, as we have incorporated many of the requested changes. We also respond to specific comments below.
>
> > W1 & RC1: On the presentation of the theoretical analysis in Sections 3 and 6
>
> Thank you for the suggestion! When writing the paper, we believed that keeping some of the analysis in Sections 3 and 6 separate was useful, as part of the additional analysis in Section 6 is not key to understanding the key points relevant to our discussion of the deficiencies of the standard DPO objective, and how it motivates our approach. Furthermore, some of the analysis (Propositions 3, 4, and 7) pertain to our proposed solutions, which are not yet introduced in Section 3. Nevertheless, we do agree with the reviewer that the analysis of Propositions 5 and 6 do fit nicely in Section 3, and have moved them there in the updated draft.
>
> > W2: On empirical results related to probability mass being moved away from preferred solutions
>
> We have included an analysis of how pessimism mitigates the reward collapse on the preferred responses in the preference dataset in our updated draft. Specifically, we test the effect of pessimism in both DPO (DPO vs p-DPO) and distilled DPO (d-DPO vs dp-DPO). See Figure B.1 of the Appendix.
>
> Note that a subtle point arises from the distinction between regularizing to the reference policy output distribution versus regularizing to the preference data distribution, which are only (asymptotically) identical if preferences are annotated on a sample from the reference policy (in our initial experiments we had found slightly better results overall when regularizing to the reference policy output distribution). In the new Appendix we report results with respect to both distributions, showing that pessimism (a) mitigates the decrease in probability of preferred and dispreferred preference annotations, despite this data not being used in the regularizers (left-most subplots), and (b) mitigates the increase in KL divergence with respect to the reference distribution (third subplot from left), as expected due to the additional regularization term. As we argue in the paper (see also Azar et. al., 2023), the $\beta$ hyperparameter of DPO does not effectively regularize this KL distribution because the implicit DPO reward model assigns infinite-magnitude rewards.
>
> > W3 & RC4: On comparisons to other reward model-based approaches
>
> To our knowledge, no other reward model-based approaches (that take a different form from our basic squared distillation loss—c.f. our discussion of REBEL with Reviewer RV9o) have been considered in the offline setting. Since our analysis and experimentation is focused solely on the offline setting, it is out of scope of this paper to compare to online methods that also use reward models (by necessity, as they are online).
>
> > RC2: Interpreting the Y-axis of Figure 2
>
> Thank you for this suggestion! We have updated the caption. The average reward for the base SFT model according to the “oracle” reward model used is around  -1. Therefore a reward of -0.5 at $\rho = 0.2$ reflects a **positive** improvement in the learned policy over SFT despite the biased preference data (and an improvement that is 1.3-2x that of IPO and DPO).
>
> > RC3: On additional datasets used to study distribution shift
>
> Thank you for sharing the reference to Lin et. al! The setting in that dataset is different from what we focus on in this paper. Their work is on evaluating the out-of-distribution accuracies of reparameterized reward models obtained via DPO vs. explicit reward models trained with standard MLE, and suggests that the explicit reward models generalize better on out-of-domain prompts or generations. Similar findings have also been reported elsewhere in the literature, see, e.g., Tang et. al. 2024. The implication is which kind of reward model to use for online preference learning, such as in iterative DPO or PPO. Of course, the implicit DPO reward model and the DPO policy are connected (i.e., the implicit reward model is the reward that the DPO policy maximizes), and understanding how the generalization abilities of the implicit reward model when used as a classifier relates to the quality of the generative policy is an interesting area of future work.
>
> [1] Tang et. al. 2024. https://arxiv.org/abs/2405.08448.

---

> > ### Comment · Reviewer_QDvm · 2025-01-02
> >
> > Many thanks for the authors for their rebuttal, paper updates and clarifications above. I want to note here that I have gone through the paper and am happy with the updates made to the paper. I have no further questions for the authors at this point.
> >
> > I look forward to moving the review process along and helping the AE in my capacity as a reviewer to complete the process here.

---

### Review · Reviewer_RV8o · 2024-12-20

**Summary Of Contributions:**

The paper proposes a hybrid approach to fix the issues underlying DPO style methods. To combat the issue of arbitrary scale of reward functions learned by DPO and potentially OOD transfer of probability mass, authors propose to learn a seperate set of rewards and use it to distill policies. Furthermore, to make their method robust they use a pessimistic reward optimization procedure.

**Audience:**

No

**Claims And Evidence:**

Yes

**Requested Changes:**

I think the contribution of this paper is investigating the already published REBEL objective for the offline setting and figuring out ways to improve offline RLHF. This contribution is perfectely reasonable but the paper currently has over claims on multiple parts and from reading seems like reward model distillation + pessimism along with theoretical analysis is the novelty of the paper. Assumi that the novelty is investigation of REBEL in offline setting, a single dataset experiments with 1 seed seems insufficient to conclude the novelty of the method.

**Strengths And Weaknesses:**

Strengths:
1. The paper addresses a known issue with DPO class of algorithms. The paper proposed a new pessismitic reward model distillation approach to handle the known issues of probability mass transfer in DPO.


Weaknesses:
1. In the abstract as well as further places in the text, the authors mention that they "analyze the phenomena of probability of preferred generations going to zero in DPO". It seems like prior work has shown this both theoretically and experimentally. Refer [1]. Proposition 1 and Corollary 1 of this paper follows from Proposition 1 in the paper [1].I think it might be more accurate to cite the relevant sources and make clear that this is not the contribution of this paper.

2. The authors propose the algorithm in section 4.1 of reward model distillation. This algorithm matches exactly to the REBEL algorithm [2] accepted in NeurIPS 2024. Refer to equation 5 in their paper vs the equation 7 in this paper. I fail to understand if the authors are claiming this as their contribution or this is a interlude to the pessimistic approach they go towards in the next section. The theorems derived in this section also follow from the proofs of paper[2] where they show the reward model distillation approach is a principled way to solve the RLHF objective.

3. While the [2] paper is purely in the online setting and I see the difference that authors aim to use this technique in the offline setting but the offline setting seems to be just one iteration of the online setting.

4. For the implementation of pessimistic model distillation is the reward pessimism enforced per state (the worst reward per prompt) or it is the worst reward function over all prompts in an expected sense? The proofs indicate later but the empirical objective indicates former.

5. Empirical results: The experiments on TLDR show a very slight improvement over IPO and it seems insufficient to conclude that this method actually results in consistent improvements. On the helpfulness dataset, other offline algorithms are not evaluated. The issues that motivate this paper plague all offline alignment algorithms (DPO, IPO, etc) as shown in [1] and the fixes here should result in consistent improvement over other offline baselines as well.


[1]: Rafailov, Rafael, et al. "Scaling laws for reward model overoptimization in direct alignment algorithms." arXiv preprint arXiv:2406.02900 (2024).
[2]: Gao, Zhaolin, et al. "Rebel: Reinforcement learning via regressing relative rewards." arXiv preprint arXiv:2404.16767 (2024).

---

> ### Author Response · Authors · 2024-12-25
>
> We thank the reviewer for all their time spent reviewing this paper, and for their helpful and constructive comments!
>
> > W1: On Rafailov et. al., 2024.
>
> Thank you for bringing this work to our attention! Rafailov et. al. do address a similar phenomenon, yet our theoretical results actually highlight different aspects of the DPO loss optima:
> - Prop. 1 of Rafailov et. al. observes that if the preference dataset does not cover the full output space of the LM, then there are an infinite number of minima to the DPO loss that place weight on “out of distribution” responses y that are not in the preference data.
> - In contrast, our Prop. 1 and Cor. 1 state that if the set of preferred / dispreferred responses is disjoint, then the minima of the DPO objective (a) “overfit” and must set the probability of all dispreferred responses to 0, and (b) are otherwise underconstrained in terms of what responses receive non-zero mass. Therefore, our result **includes** placing non-zero mass on potentially many of the OOD responses that Prop. 1 of Rafailov et. al. discusses. However, while Prop. 1 of Rafailov et. al. notes that OOD responses can receive **non-zero mass**, it does not show that the preferred responses can also receive **near zero mass**, which is what we demonstrate.
> - Our theoretical analysis in Sec. 7 (now partly moved to 3) on characterizing the optima of DPO and **why** it can drive $\pi(y^w)$ to 0 also extends the analysis well beyond that of Rafailov et. al.
>
> We have updated our Related Work.
>
> > W2 & W3: On Gao et. al., 2024.
>
> Thank you for also pointing out the REBEL algorithm in Gao et. al., 2024! There are also a few key differences between our work and REBEL:
>
> - Our work focuses on the offline setting and the distinct theoretical and empirical challenges it brings over online settings. REBEL is an online algorithm, and evaluated only in online settings.
> - Moreover, while the two squared losses are equivalent, coming from the offline setting, our motivation is quite different: our point is that (a) using trained reward models in offline settings still brings a significant amount of benefit in terms of optimization and (b) still admits a simple and efficient DPO-like objective. This is different in argument & spirit from REBEL, which being an online algorithm, must use a reward model, and is compared to online algorithms which also must use reward models.
> - Our distillation objective also serves as an introduction & interlude to our offline pessimistic variants, which is not at all covered in REBEL.
>
> We have updated our Related Work.
>
> > W4: On pointwise vs. populationwise pessimism
>
> Our pessimistic objective optimizes the worst case reward in expectation. This is true for both our theoretical analysis and empirical objective. The pointwise minimum over target reward models in the empirical objective corresponds to our Lagrangian-style penalty for softly restricting our pessimistic reward model set to the desired target one, $\mathcal{S}$, which we motivate in Prop. 2.
>
> > W5: On the empirical comparison to IPO
>
> We respectfully disagree that the improvement over IPO is minor: in Figure 2, our method outperforms IPO at **all** length biases, with **substantial** improvement over IPO for all length biases below 0.5. In terms of robustness, we believe this is a very important result.
>
> > On novelty & audience
>
> We would like to reiterate that this work is not a reinvestigation of REBEL:
> 1. The motivation for our work is completely different, and arises from a theoretical investigation of offline DPO (which also, again, we believe is novel and complementary to that of Rafailov et. al.).
> 2. While the most basic form of our distillation objective indeed takes the same squared loss form as REBEL, there is nonetheless a significant element of novelty in that (a) it is shown here to easily extend to offline pessimism, and (b) it highlights the (not necessarily intuitive) empirical and theoretical value and practicality of using real-valued reward model in the offline setting as well as the online setting.
>
> Finally, in light of the TMLR guidelines,
>
> > “Crucially, it should not be used as a reason to reject work that isn't considered “significant” or “impactful” because it isn't achieving a new state-of-the-art on some benchmark. Nor should it form the basis for rejecting work on a method considered not “novel enough”, as novelty of the studied method is not a necessary criteria for acceptance [...]”
>
> We argue that, despite the similarity to these papers (which appeared at NeurIPS post TMLR submission), our paper offers valuable new insights, connections, and empirical results of both theoretical and practical significance to the offline RLHF community. Importantly, the congruence of our findings with the independent REBEL work—where both highlight the value of reward difference regression in offline (ours) *and* online (REBEL) settings—strongly signals the promise of this research direction for the community.

---

> > ### Comment · Reviewer_RV8o · 2025-01-04
> >
> > > However, while Prop. 1 of Rafailov et. al. notes that OOD responses can receive non-zero mass, it does not show that the preferred responses can also receive near zero mass, which is what we demonstrate.
> >
> > Prop. 1 of Rafailov et. al. shows that there are infinite solutions that place mass OOD and in their set also contains setting where preferred responses receives zero mass. In fact, their figure 10 also demonstrates empirically that this can happen in a simplified setting of Tree MDP. I still believe the paper should correct its claim regarding presenting the DPO behavior as a new finding of this work.
> >
> > > Moreover, while the two squared losses are equivalent, coming from the offline setting, our motivation is quite different: our point is that (a) using trained reward models in offline settings still brings a significant amount of benefit in terms of optimization and (b) still admits a simple and efficient DPO-like objective. This is different in argument & spirit from REBEL, which being an online algorithm, must use a reward model, and is compared to online algorithms which also must use reward models
> >
> > While I agree with the interesting notion of testing the idea in the offline setting, the paper still presents the section as it is first introduced as part of the work. The paper should make proper citation to REBEL and not attempt to present the distillation objective as novel in the methods section.
> >
> >
> > > Finally, in light of the TMLR guidelines,
> >
> > >“Crucially, it should not be used as a reason to reject work that isn't considered “significant” or “impactful” because it isn't achieving a new state-of-the-art on some benchmark. Nor should it form the basis for rejecting work on a method considered not “novel enough”, as novelty of the studied method is not a necessary criteria for acceptance [...]”
> >
> > > We argue that, despite the similarity to these papers (which appeared at NeurIPS post TMLR submission), our paper offers valuable new insights, connections, and empirical results of both theoretical and practical significance to the offline RLHF community. Importantly, the congruence of our findings with the independent REBEL work—where both highlight the value of reward difference regression in offline (ours) and online (REBEL) settings—strongly signals the promise of this research direction for the community.
> >
> > I appreciate the authors reiterating the guidelines of TMLR. The papers mentioned have been on arXiv (REBEL 25 Apr 2024 and Scaling Laws 5 Jun 2024) much before the TMLR submission period. Attempting to highlight results as new despite existing goes against the core values of the research community. My comments do not reduce the value of the paper for the fact that the paper is investigating an offline setting as opposed to an online setting that REBEL investigated with a similar objective with some modifications (the pessimistic setting). My comments are more towards the proper citation of what prior methods have done and clearly delineating what this paper is contributing; something which is still missing from the paper. TMLR guidelines do not encourage regurgitating ideas from previous papers as new.

---

> > > ### Author Response · Authors · 2025-01-04
> > >
> > > After taking a more careful look at Prop. 1 of Rafailov (and Hejna et al., which it builds on), we agree with the reviewer that this claim can indeed be inferred from their proof, even though it is not immediately apparent from the statement of the proposition itself. We thank the reviewer for pointing this out! We have now updated our draft to point out that Rafailov et. al. make the same observation theoretically and empirically in independent work.

---

> > > > ### Comment · Reviewer_RV8o · 2025-01-14
> > > >
> > > > I appreciate the author's response.
> > > >
> > > > Can the authors:
> > > > 1. Modify section 4 to clearly indicate the connections and relations to Rafailov et al. It is okay to write that it is concurrent work.
> > > > 2. Modify section 5.1 to to clearly indicate the connections and relations to REBEL. It is okay to write that it is concurrent work.

---

> > > > > ### Author Response · Authors · 2025-01-16
> > > > >
> > > > > Thanks again for the constructive feedback. We have uploaded a new draft with modifications to Sections 4 and 5.1 that more clearly emphasizes the connections to the discussed work.

---

### Author Response · Authors · 2024-12-31
**Thank you to reviewers and a request for comments**

Dear Reviewers,

Thank you for your constructive review! We have provided our individual responses, and also uploaded an improved draft that addresses the requested changes that were made by reviewers. Let us know if there is anything else that would benefit from further discussion.

Authors

---

### Decision · Action_Editor_Ey8R · 2025-02-04

**Recommendation:** Accept as is

**Comment:**

This paper presents a method that integrates reward model distillation with pessimistic optimization to address limitations in DPO. It aims to mitigate probability mass transfer issues and improve stability in preference learning. The experiments show the superioity of the proposed method over a few baselines.
    A major concern from the reviewer QDvm is whether the relation to Gao et al. is clearly clarified. The authors explained that the submission focused on the offline setting and is quite different from Gao et al. After checking related works, I think the submission has a quite different motivation from Gao et al. and these two works can be viewed as concurrent works. In addition, another mentioned work Rafailov et. al. shall also be viewed as concurrent work. I think the submission can be accepted if the relation to these two works is clarified. The authors further modified the discussions on the relations to related works, and the reviewer QDvm is satisfied with the modifications.
    Overall, I think the paper presents a meaningful contribution to RLHF and preference optimization. I recommend acceptance.

**Audience:**

The research is relevant to the TMLR audience, particularly those interested in reinforcement learning from human feedback (RLHF), preference optimization, and alignment. The topic aligns well with recent advancements in RLHF, making it a timely contribution.

**Claims And Evidence:**

The paper presents an approach to improving Direct Preference Optimization (DPO) by incorporating reward model distillation and a pessimistic optimization technique to address an issue related to probability mass transfer. The presented results verified that the proposed method improves the performance.
    As one reviewer asked, to justify the claim that probability mass moved away from preferred solutions, more empirical results are needed. The authors added experiments to validate that pessimism mitigates reward collapse on the preferred solutions in the updated manuscript. Thus I think the claims in the paper are well justified.